

# Determining Minnesota bee species' distributions and phenologies with the help of participatory science

Colleen D. Satyshur[1], Elaine C. Evans[2,3], Britt M. Forsberg[2], Thea A. Evans[1] and Robert Blair[4]

[1] Department of Ecology, Evolution and Behavior, University of Minnesota, St. Paul, MN, United States of America
[2] University of Minnesota Extension, University of Minnesota, St. Paul, MN, United States of America
[3] Department of Entomology, University of Minnesota, St. Paul, MN, United States of America
[4] Department of Fisheries, Wildlife and Conservation Biology, University of Minnesota, St. Paul, MN, United States of America

## ABSTRACT

The Minnesota Bee Atlas project contributed new information about bee distributions, phenologies, and community structure by mobilizing participatory science volunteers to document bees statewide. Volunteers submitted iNaturalist (©2016 California Academy of Sciences) photograph observations, monitored nest-traps for tunnel-nesting bees, and conducted roadside observational bumble bee surveys. By pairing research scientists and participatory science volunteers, we overcame geographic and temporal challenges to document the presence, phenologies, and abundances of species. Minnesota Bee Atlas project observations included new state records for *Megachile inimica*, *Megachile frugalis*, *Megachile sculpturalis*, *Osmia georgica*, *Stelis permaculata*, and *Bombus nevadensis*, nesting phenology for 17 species, a new documentation of bivoltinism for *Megachile relativa* in Minnesota, and over 500 observations of the endangered species *Bombus affinis*. We also expanded known ranges for 16 bee species compared with specimens available from the University of Minnesota (UMN) Insect Collection. Surveys with standardized effort across the state found ecological province associations for six tunnel-nesting species and lower bumble bee abundance in the Prairie Parkland ecological province than the Laurentian Mixed Forest or Eastern Broadleaf Forest ecological provinces, indicating potential benefit of a focus on bumble bee habitat management in the Prairie Parkland. Landcover analysis found associations for four tunnel-nesting species, as well as a possible association of *B. affinis* with developed areas. These data can inform management decisions affecting pollinator conservation and recovery of endangered species. By engaging over 2,500 project volunteers and other iNaturalist users, we also promoted conservation action for pollinators through our educational programs and interactions.

Corresponding author
Colleen D. Satyshur,
csatyshu@umn.edu

## INTRODUCTION

While bees are widely recognized for their important role in food security and the maintenance of ecological integrity (*Klein et al., 2007*; *Ollerton, Winfree & Tarrant, 2011*), the monitoring and baseline information necessary for regional bee conservation is often missing (*Cardoso et al., 2011*; *Lebuhn et al., 2013*). Without such data on species distributions, habitat associations, and phenology, it is difficult to understand if or how bee communities are changing or how to enact conservation practices. Knowing species distributions and estimates of abundance can help prioritize management and conservation efforts (*Cardoso et al., 2011*). For example, species with small geographic distributions are at higher risk of extinction (*Gaston & Fuller, 2009*). Habitat associations are also important because bees are often closely tied to plant communities (*Potts et al., 2003*; *Sheffield & Heron, 2019*) and habitat needs such as nest sites (*Potts et al., 2003*; *Harmon-Threatt, 2020*). In addition, establishing phenology baselines is important to understanding the ecological role of bee species and how climate change impacts ecosystems now and in the future (*Burkle, Marlin & Knight, 2013*; *Ogilvie & Forrest, 2017*).

The importance of baseline information has led to calls for developing national survey and monitoring programs to support state-based pollinator conservation plans (*Woodard et al., 2020*). While recent efforts list over 500 bee species in Minnesota (*Portman et al., 2023*), the distribution, population, and life history traits such as nesting phenology, often remain unknown. There are four distinct ecological provinces in the state: Prairie Parklands (PP), Tallgrass Aspen Parklands (TAP), Eastern Broadleaf Forest (EBF), and Laurentian Mixed Forest (LMF) (*Minnesota Department of Natural Resources, 2023*). The effort and funds required to survey these ecologically different areas of the state for insect pollinators are a challenge. Additionally, commonly used methods for studying insects require extensive specimen collection and taxonomic expertise for species-level identification for most groups, which can also be expensive (*Woodard et al., 2020*).

Inviting the public to participate in scientific research can help overcome geographic and temporal challenges of bee monitoring. Here we use the term participatory science (sometimes called citizen science or community science) to indicate volunteer participants who are not monetarily compensated. Participatory science contributions can provide complementary and widespread records across locations and time, contributing observations earlier in the season and of a significantly broader distribution than professional datasets alone (*Van der Wal et al., 2015*; *Soroye, Ahmed & Kerr, 2018*; *Dubaić et al., 2022*). Structured participatory science projects in North America and Europe have also produced data of sufficient quality to be used in monitoring, conservation, and management (*Kremen, Ullman & Thorp, 2011*; *Appenfeller, Lloyd & Szendrei, 2020*; *Koffler et al., 2021*), documented natural history traits such as nesting and seasonality (*Lye et al., 2012*; *Maher, Manco & Ings, 2019*; *Olsen et al., 2020*) and increased conservation action (*Ganzevoort & Van den Born, 2021*; *Griffin et al., 2021*).

In this study, we leveraged the power of participatory science to investigate bee distribution, nesting phenology, and community structure across the state of Minnesota in the U.S. We engaged volunteers in three tiers of sampling rigor: (1) casual observations of

all bee species using the mobile app and website iNaturalist.org (©California Academy of Sciences 2016), (2) nest-trap surveys of tunnel-nesting bees, and (3) observational bumble bee surveys. The three tiers of sampling rigor represent increasing levels of volunteer training and commitment and yielded different data types. The iNaturalist observations required minimal training and flexible volunteer time commitment. While not appropriate for all bee species, the use of crowd-sourced identifications provided presence data for bee species amenable to identification from photographs, particularly bumble bees. The nest-trap surveys required more training and a season-long commitment from volunteers. They provided distribution, ecological association, nesting phenology, and nesting biology data for a subset of bees that are often not well represented in other survey methods (*Westphal et al., 2008*; *Staab et al., 2018*). Volunteers who worked on bumble bee surveys had in-depth training on bumble bee identification and sampling methods and committed to a more time-intensive sampling protocol. Bumble bee surveys used equal sampling effort across observations to provide abundance and distribution data, as well as indication of habitat associations. Together, these data will inform statewide pollinator conservation plans and contribute to baseline assessments for evaluating the status of pollinators in Minnesota in the future.

## MATERIALS & METHODS

The Minnesota Bee Atlas participatory science project operated between 2016-2020. We recruited volunteers statewide (Fig. 1) by advertising to local volunteer groups and conservation organizations, on social media, and through University of Minnesota web pages. Volunteers had various affiliations including the Minnesota Master Naturalist program, Minnesota Department of Natural Resources Scientific and Natural Area stewards, Environmental Learning Centers, nature centers, county natural resource departments, Soil and Water Conservation Districts, native plant nurseries, and federal agencies including the U.S. Forest Service and the U.S. Fish and Wildlife Service. Approximately 150 volunteers engaged with project staff and participated in one of the three protocol areas each field season. As of March 2021, 2,300 users submitted observations of bees in Minnesota to iNaturalist, some of whom specifically contributed to the MN Bee Atlas project and many of whom submitted bee observations that the portal automatically added to the project. Over 1,000 users contributed identifications to MN Bee Atlas iNaturalist records.

### iNaturalist

The broadest and simplest level of participation relied on the mobile app and website iNaturalist. This global public biodiversity portal enables individuals to upload locations and evidence of living things, including photos or recordings, which are then identified by the observer, other users, or an algorithmic suggestion based on existing research-grade observations. Each identification is qualified based on a data validation system and considered research-grade if an observation is not of a captive or cultivated species, has a date, photo and location, and two-thirds of users agree on genus and species-level identification. This is not foolproof, as there are no required credentials to add

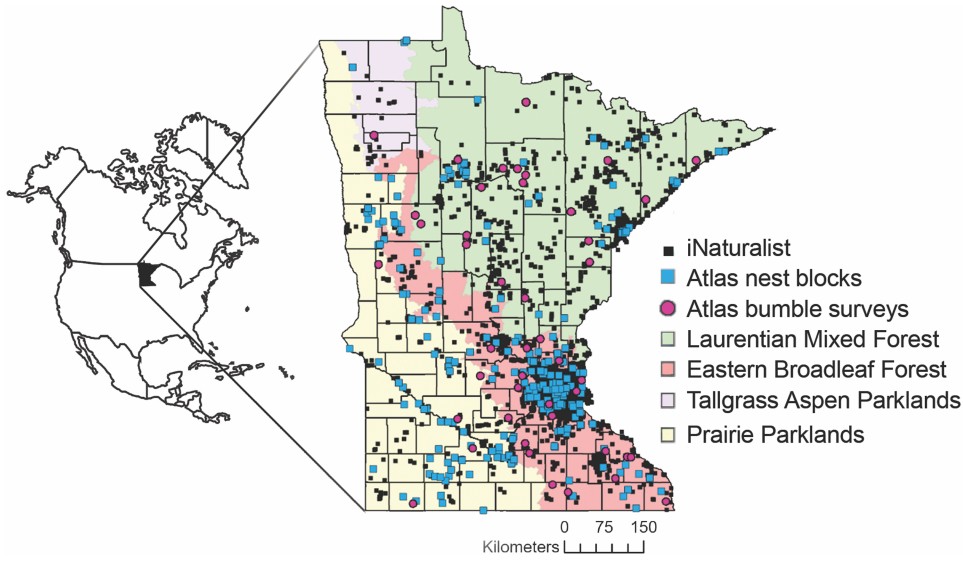

**Figure 1** **Locations of Minnesota Bee Atlas observations.** Observations include research grade iNaturalist observations of bees between 29 July 2005 and 9 March 2021, nest traps and stem bundles monitored from 2016 to 2019, and bumble bee routes surveyed from 2016 to 2020. Observations took place across Minnesota's four ecological provinces. Maps in this study were created using Esri ArcGIS Online with MN DNR layer: Ecological Sections of Minnesota; and Esri layers: United States State Boundaries 2018, World Ocean Reference (English), Ocean/World_Ocean_Base. Provinces and Territories of Canada.

identification. However, there are many knowledgeable iNaturalist users, both professionals and experienced enthusiasts, who spend time identifying iNaturalist observations from others and are integral to the creation of research-quality data. The quality of identification typically grows over time as additional users join the platform and as additional identification experts participate. We examined a subset of research-grade observations from genera that are difficult to identify to species (*i.e., Andrena, Lasioglossum, Nomada*). These records were verified by expert bee taxonomists, including John Ascher, Jason Gibbs, and Zach Portman. Once identifications reach research-grade, records feed into databases such as GBIF (http://www.gbif.org). We trained 338 participants who attended workshops to add bee observations to iNaturalist and to identify bees to groups, usually family. Most workshop participants added observations to iNaturalist, with a small percentage becoming regular contributors or identifiers.

## Tunnel-nesting bees

Tunnel-nesting bees nest in above-ground tunnels in wood or plant stems. Each female builds her own nest by constructing a series of compartments. In each compartment she stores pollen and nectar and lays a single egg. When the nest is complete, she plugs the tunnel entrance, leaving the young to develop on their own. Different species use different materials for nest plugs. Many species will also nest in artificial nest-traps which can be used as a survey method. In this study, participants hung and monitored wood nest-traps in semi-natural habitats on private or public lands from April to October. Nest-trap design and nest plug descriptions were adapted from The Bees' Needs (*Rose, Scott & Bowers, 2015*;
V. Scott, pers. comm., 2016). We drilled five tunnels of six different diameters (3.18 mm, 4.76 mm, 6.35 mm, 7.94 mm, 9.53 mm, and 11.11 mm) into blocks of untreated pine or Douglas fir with a cedar shingle roof (Appendix S1). We use the term "nest" to mean a tunnel that produced a particular bee species. Different species sometimes build sequential nests in the same tunnel. Occasionally, different individuals from the same species may nest within the same tunnel, but for this study we assumed individuals of the same species within a tunnel were from the same mother.

With the goal of surveying the whole state, we actively recruited volunteers to hang nest-traps in rural areas and in areas with less existing data. Volunteers attended in-person or online training and received a written instruction manual with photographs of different plug materials. They placed nest-traps in a semi-sunny location facing east or south at a height of 1 to 2 m, with the flexibility to find a mounting site that fit their habitat. Volunteers were instructed to report plugged tunnels or other nest evidence every 2–3 weeks *via* the project web page. Bee Atlas staff provided feedback on observations *via* email and newsletters. In 2016, 2017, and 2018, we sent out 120, 129, and 141 nest-traps respectively and 116, 127, and 140 were returned, respectively, for a return rate of 98%. Nest-traps were distributed across 60 of the 87 Minnesota counties and all four ecological provinces, including 69 in the LMF, 224 in the EBF, 87 in the PP, and two in the TAP ecological provinces (Fig. 1). The Minnesota Department of Natural Resources approved research permit numbers 2016-29, 2016- 4R, 201705, 2017-9R, 201822, and 2018-15R for nest-traps placed in State Parks, State Forests, Scientific and Natural Areas and Wildlife Management Areas.

We received one homemade nest-trap bundle made from *Phragmites* stems from one volunteer in Brown County each year between 2016 and 2018. In 2019, the final year, we sent 11 additional nest bundles made with hollow or pithy plant stems to selected volunteers to observe nesting with different natural substrates. We made each bundle from stems of one of six native plant species; *Asclepias incarnata*, *Silphium perfoliatum*, *Arnoglossum atriplicifolium*, *Helianthus giganteus*, *Vernonia fasciculata*, or *Liatris ligulistylis*, and placed bundles inside a plastic sleeve with an overhanging roof made from a 64 oz (1.89 liter) beverage bottle. We sealed the backs of the stems with cotton balls and latex. The number of stems per bundle varied due to the size differences between stems. Monitoring protocols were like those used for wood nest-traps.

In the late fall, volunteers returned nest-traps and stem bundles to the University of Minnesota for overwintering and rearing in a temperature-controlled growth chamber as described in *Satyshur et al. (2020)*. After a four-month period at 5 °C, we stimulated emergence by increasing the temperature in steps to a high of 30 °C. We covered each nest-trap tunnel entrance with test tubes and removed emerging insects daily. Bundles were reared in bags. Some bees appeared to have already emerged by fall 2016, so in 2017 and 2018, we swapped out a few nest-traps with similar plugs in mid-summer and reared them in the lab at ambient temperature. We (CS, TE) identified bees to species using keys (*Sandhouse, 1939*; *Mitchell, 1962*; *Sheffield et al., 2011*; *Arduser, 2018*; *Andrus, Droege & Griswold, 2020a*; *Andrus, Droege & Griswold, 2020b*; *Andrus, Droege & Griswold, 2020c*; *Griswold et al., 2020*; *Nelson & Droege, 2020*; *Nelson & Droege, 2020b*; *Orr et al., 2020*) and

comparisons with previously identified specimens. Jason Gibbs, Michael Orr, Ryan Oram, and Sam Droege confirmed identification of more difficult specimens. We identified wasps using keys (*Gibson, Huber & Woolley, 1997*; *Triplehorn, 2005*; *Heraty, 2008*). John Lumen identified all Ichneumonidae and provided consultation on Chalcidoidea. *Kocourekia* cf. *debilis* was identified to species using *Cao et al. (2017)* and verified by Jorge González and Mike Gates. We deposited voucher specimens in the UMN Insect Collection. We included locations of specimens in the UMN Insect Collection database when mapping species distributions. Many UMN Insect Collection specimens did not have latitude or longitude associated with their records. In such cases, we used the location description to estimate the most accurate position possible. We chose the approximate center of geographic areas such as cities and state parks. If only county location was available, we placed the specimen in the approximate center of the county and identified the records as such.

We examined nesting phenology using volunteer-submitted nest plug observations. For each nest tunnel that produced bee offspring, project staff evaluated observations and assigned a quality value based on clarity and frequency of observations. Higher values were assigned if the full plug observation was clear and consistently observed following formation and if observations were four weeks apart or less. Nest tunnels with high or medium quality values were used in phenological estimations, with 65.1% of observations meeting those criteria. Because volunteers checked approximately every two to three weeks, we could determine that nest completion occurred in the interval between the last date that the volunteer recorded an empty tunnel and the first date with a complete nest plug. We assumed nests were equally likely to be completed on any particular day in an interval and assigned each day an equal probability. We summed these probabilities over all nests with sufficient quality observations and determined the median date. We also calculated the 0.25 and 0.75 quartile values, which bound a central period when nests were most likely completed.

## Bumble bees

We trained volunteers in survey methods and skills to distinguish bumble bees from other insects, determine sex, identify readily distinguishable bumble bee species, and photograph bumble bees to enable identification. Based on regional collections, we estimated that 90% of observations would be readily distinguishable species (*Bombus impatiens* Cresson, 1863, *Bombus bimaculatus* Cresson, 1863, *Bombus griseocollis* (De Geer, 1773), or *Bombus ternarius* Say, 1837). We adapted survey methods from previous state-wide bumble bee surveys that used lethal collection methods (*Golick & Ellis, 2006*; *McFarland, Richardson & Zahendra, 2015*; *Richardson et al., 2019*). Due to volunteer preferences and the presence of federally protected *Bombus affinis* Cresson, 1863, we used observational data instead of specimen collections. Forty-four volunteers observed bees at five stops along 39.5-kilometer routes between 10 a.m. and 6 p.m. on days with little or no precipitation, temperatures greater than 15.6 °C, and wind speeds less than 32.2 kph. We requested volunteers survey along their route three times each year, between late June and mid-August with at least two weeks between visits. Volunteers surveyed 45 of 90 available routes between 2016 and 2020, with 37 routes with three completed route runs per year, and 17 routes surveyed for

**Table 1 Bumble bee survey routes.** Volunteers adopted routes and completed surveys (three route runs with five 10-minute observations per route run) along routes between 2016 and 2020 across the Prairie Parkland (PP), Laurentian Mixed Forest (LMF), and Eastern Broadleaf Forest (EBF). Only one route was adopted in the Tallgrass Aspen Parklands province. This route is included in totals for the Prairie Parkland for routes adopted but did not have any completed surveys. Land cover was determined within 2 km of routes using the 2016 National Land Cover Database (NLCD) (*Dewitz, 2019*) verified by examining aerial photographs.

| Ecological province | Routes adopted | Total complete surveys | Surveys in 2016, 2017, 2018, 2019, or 2020 | Dominant, secondary land covers |
|---|---|---|---|---|
| Prairie Parkland | 6 | 6 on 4 routes | 0, 2, 1, 2, 1 | crops, wetlands |
| Laurentian Mixed Forest | 18 | 28 on 14 routes | 2, 6, 6, 6, 8 | wetlands, forest |
| Eastern Broadleaf Forest | 21 | 45 on 19 routes | 5, 8, 10, 11, 11 | crops, forest |
| Overall | 45 | 79 on 37 routes | 7, 16, 17, 19, 20 | |

three or more years (Fig. 1, Table 1). Routes were based on established North American Breeding Bird Survey routes (*USGS Patuxent Wildlife Research Center, 2017*) because of their accessibility and systematic spread across different ecological areas. For analysis, we combined the single route from the TAP ecological province with routes from the PP ecological province due to the low sample size in this province and ecological similarity. Volunteers chose five stops along a route by finding flower patches with bee activity located at least 1.61 km (1 mile) from each other. On average, survey stops were 5.23 kilometers apart from each other. Volunteers examined flower patches within 150 m of the survey stop, collecting bumble bees from flowers into jars for ten minutes of collecting time and noting the flower's identity. Volunteers placed bees in coolers with ice to avoid risk of bees overheating and to ease photography. Volunteers counted and released readily identifiable individuals and photographed a subset of bees including all bees that were not readily identifiable, all bees belonging to the subgenus *Psithyrus* other than *Bombus citrinus* (Smith, 1854), and all individuals of conservation concern (*B. affinis*, *Bombus terricola* Cresson, 1863, *Bombus pensylvanicus* (De Geer, 1773)) as listed by the International Union for the Conservation of Nature (*Hatfield et al., 2015*). Volunteers submitted data through the Bee Atlas website. We (EE) verified identifications for all photo-specimens. Most specimens (89%) were identified by volunteers, with 10% of specimens verified with photographs, and 1% unverifiable due to poor photo quality. Two species, *Bombus vagans* Smith, 1854 and *Bombus sandersoni* Franklin, 1913, were grouped because most observations did not include identifying features that enabled separation of these two closely related species from each other.

## Statistical analysis

We used R (*R Core Team, 2022*) and Rstudio (*R Studio Team, 2022*) for all statistical analyses. We examined differences among ecological provinces for tunnel-nesting bees and bumble bees using generalized linear mixed-effect models in the glmmTMB R package (*Brooks et al., 2017*) with post-hoc comparisons of estimated marginal means using the R package emmeans (*Lenth et al., 2023*). We checked all model residuals for overdispersion and heteroscedasticity. We compared overall frequency of tunnel use by nesting bees across

the LMF, EBF, and PP with a negative-binomial model to account for the high numbers of zeros in the data. We did not include the TAP since there were only two nest-traps in that province. We also used negative binomial distribution to model annual nest counts per nest-trap per species by ecological province, with year and location as random effects. The location variable grouped nest-traps that were within one kilometer of one another. We selected the following nest-building species for this analysis based on presence in 30 or more nest-traps (10% or more of all nest-traps): *Heriades carinata* Cresson, 1864, *Megachile campanulae* (Robertson, 1903), *Megachile pugnata* Say, 1837, *Megachile relativa* Cresson, 1878, *Megachile rotundata* (Fabricius, 1787), *Osmia lignaria* Say, 1837, *Osmia pumila* Cresson, 1864, and *Osmia tersula* Cockerell, 1912. We did not include parasitic species in this analysis due to their correlation with their host species. *Megachile campanulae* and *O. pumila* were not recorded by nest-traps in the LMF and were analyzed for PP and EBF only. We created models for bumble bees with log-transformed abundance of bumble bees per route per year as the response variable and ecological province as the predictor with year and route as random effects. After preliminary analysis, we changed year from a random to a fixed effect due to singularity. We limited data to include only routes with three completed route runs (a set of five 10-minute observations) within a year, which equaled 150 min of survey time, to ensure equal sampling across routes. We included all observations of bumble bees.

We summarized land cover in areas surrounding nest-traps and bumble bee routes using the 2016 National Land Cover Database (NLCD) (*Dewitz, 2019*). We verified land-cover categories by randomly spot checking against aerial photographs across approximately 25% of surveyed areas and checking all areas characterized as barren, as that NLCD category can have a higher error rate (*Hollister et al., 2004*). Land use surrounding one nest-trap that was near the border with Canada was supplemented with visual assessment from aerial photos because NLCD data was only available for half of the buffer area surrounding the nest-trap site. For tunnel-nesting bees, we examined land cover within a radius of 250 m of nest-traps (*Gathmann & Tscharntke, 2002*; *Steffan-Dewenter et al., 2002*). For bumble bees, we examined land cover within a 2 km radius of the center of all bumble bee survey stops and summed them for each route (*Hagen, Wikelski & Kissling, 2011*; *Rao & Strange, 2012*). We simplified NLCD land-cover classes to groupings that we consider to be biologically relevant to bee distribution (*Holzschuh, Steffan-Dewenter & Tscharntke, 2010*; *Westerfelt, Weslien & Widenfalk, 2018*; *Lanterman et al., 2019*). We combined deciduous, mixed, and evergreen forest into the forested category, all developed categories into one developed category, grasslands/herbaceous and pasture/hay into the grasslands category, and woody wetlands and emergent herbaceous wetlands into the wetlands category. Crops, open water, and barren were not combined with any other categories. Land use surrounding nest-traps consisted of 28% forested, 20% grasslands, 19% developed, 12% crops, 14% wetlands, 7% open water, and 0.3% barren. Land use surrounding bumble bee route stops consisted of 26% crops, 26% forested, 24% wetlands, 11% grassland, 8% developed, 5% open water and <1% barren.

We examined the relationship of bees to land cover categories using redundancy analysis (RDA) with presence-absence for tunnel-nesting bees and constrained correspondence

analysis (CCA) with abundance for bumble bees using the vegan R package (*Oksanen et al., 2020*). For the RDA, we used forward selection using permutation tests with 1,000 permutations to select the final model. We removed the land uses crops, wetlands, open water, and barren from the final model due to lack of significance. For the CCA, we removed the variable crops due to multicollinearity (variance inflation factor > 20), the variables open water and barren due to poor correlation (intra-set correlations with axes 1, 2, or 3 < 0.4), and species accounting for less than 5% of the inertia for CCA 1 and 2 (*B. citrinus*, *Bombus insularis* (Smith, 1861), and *Bombus rufocinctus* Cresson, 1863). Significance of the overall CCA and ordination axes was determined with a Monte Carlo permutation test with 999 randomizations.

## RESULTS

### iNaturalist

People will continue contributing observations to iNaturalist indefinitely, but as of 9 March 2021, the Minnesota Bee Atlas project included 18,956 records of bees from 2,300 observers. Of these observations, 65.3% (12,384) were research-grade, slightly higher than the 60.8% rate of research-grade observations for bees worldwide in the same period (Appendix S2). Research-grade observations contained 33 genera (7 taken to subgenera) and 128 species. Of the top ten most common species identified to research-grade, nine were bumble bees (*Bombus*), and the tenth was the western honey bee (*Apis mellifera* Linnaeus 1758). Bumble bees and honey bees combined made up about 85% of the research-grade records. Other commonly recorded species included: *Agapostemon virescens* (Fabricius, 1775) (192 records), *Melissodes bimaculatus* (Lepeletier, 1825) (165), *Halictus ligatus* Say, 1837 (123), and *Megachile latimanus* Say, 1823 (118). Some bee species were notably absent in iNaturalist, particularly those in the family Halictidae (19 species were represented in iNaturalist of the 134 species known to be in Minnesota) (*Portman et al., 2023*).

The iNaturalist data include research grade records from 79 of the 87 counties in Minnesota (Fig. 1). *Bombus affinis*, the federally endangered rusty patched bumble bee, was frequently identified in iNaturalist data (over 500 observations). Public participants also documented declining bumble bee species (*B. terricola* and *B. pensylvanicus*), an introduced species (*Megachile sculpturalis* Smith, 1853), a newly documented in Minnesota species (*Bombus nevadensis* Cresson, 1874) (*Portman & Dolan, 2022*), and a rarely recorded species (*Bombus frigidus* Smith, 1854).

### Tunnel-nesting bees

From the 383 nest-traps in this study, we reared a total of 13,062 specimens, which emerged from 1,821 nest tunnels. Specimens included 3,488 solitary nest-building wasps, 1,387 parasitic wasps, and 7,123 bees from 32 species (Table 2, Appendix S3). Five bee species were cleptoparasitic, species that lay eggs in a host bee's nest. Less than one percent of bee-occupied nest tunnels were of introduced species. The bee species that occupied the greatest number of nest tunnels were *O. lignaria* (484), *Heriades carinata* (375), *O. pumila* (173), *Megachile pugnata* (151), *Megachile relativa* (132), and *Megachile campanulae* (128). The Minnesota Bee Atlas project also documented rarely collected species, including
**Table 2 Number of tunnels in trap nests that produced tunnel-nesting bee species in the four ecological provinces of Minnesota.** Between 2016 and 2019 volunteers placed 69 nest traps in the Laurentian Mixed Forest (LMF), 224 traps in the Eastern Broadleaf Forest (EBF), 87 traps in the Prairie Parkland (PP), and two traps in the Tallgrass Aspen Parkland (TAP).

| Species | Total tunnels | PP | EBF | LMF | TAP | Native/Introduced |
|---|---|---|---|---|---|---|
| *Anthophora terminalis* | 1 | | 1 | | | native |
| *Hylaeus annulatus* | 5 | 3 | | 2 | | native |
| *Hylaeus leptocephalus* | 8 | 5 | 3 | | | introduced (*Russo, 2016*) |
| *Hylaeus mesillae* (group) | 6 | 1 | 5 | | | native |
| *Hylaeus nelumbonis* | 1 | 1 | | | | native |
| *Hylaeus* sp.(*modesta*/sp.A) | 3 | | 3 | | | |
| *Hylaeus verticalis* | 4 | 2 | | 2 | | native |
| *Coelioxys alternata*[*] | 8 | 3 | 4 | 1 | | native, [*]on *M. pugnata* |
| *Coelioxys modesta*[*] | 30 | 2 | 28 | | | native, [*]on *M. campanulae* |
| *Coelioxys moesta*[*] | 11 | 1 | 2 | 8 | | native, [*]on *M. relativa* |
| *Heriades carinata* | 375 | 117 | 221 | 36 | 1 | native |
| *Heriades leavitti* | 5 | | 5 | | | native |
| *Heriades variolosa* | 22 | 18 | 4 | | | native |
| *Megachile brevis*[b] | 1 | 1 | | | | native |
| *Megachile campanulae* | 128 | 34 | 94 | | | native |
| *Megachile centuncularis* | 3 | 3 | | | | ~introduced (*Sheffield et al., 2011*) |
| *Megachile frugalis* | 1 | 1 | | | | native |
| *Megachile inermis* | 27 | 3 | 15 | 9 | | native |
| *Megachile inimica* | 5 | 2 | 3 | | | native |
| *Megachile lapponica* | 1 | | | 1 | | native |
| *Megachile mendica* | 10 | 5 | 5 | | | native |
| *Megachile pugnata* | 151 | 62 | 79 | 9 | 1 | native |
| *Megachile relativa* | 132 | 11 | 57 | 62 | 2 | native |
| *Megachile rotundata* | 36 | 14 | 20 | 2 | | introduced (*Russo, 2016*) |
| *Osmia albiventris* | 2 | | | 2 | | native |
| *Osmia georgica* | 1 | | 1 | | | native |
| *Osmia lignaria* | 484 | 43 | 245 | 195 | 1 | native |
| *Osmia pumila* | 173 | 1 | 172 | | | native |
| *Osmia tersula* | 77 | 5 | 9 | 61 | 2 | native |
| *Stelis coarctatus*[*] | 42 | 4 | 33 | 5 | | native, [*]on *H. carinata* |
| *Stelis permaculata*[*] | 3 | 3 | | | | native, [*]on *H. carinata* |
| *Hoplitis albifrons*[b] | 1 | | | 1 | | native |
| Total | 1757 | 345 | 1009 | 396 | 7 | |

**Notes.**
[*]Cleptoparasitic species: number of nests parasitized.
[b]Species only found in stem bundles.

*Megachile lapponica* Thomson, 1872 and *Hylaeus nelumbonis* (Robertson, 1890), and four species, *Megachile inimica* Cresson, 1872, *Megachile frugalis* Cresson, 1872, *Osmia georgica* Cresson, 1878 and *Stelis permaculata* Cockerell, 1898, that were new records for the state (*Satyshur et al., 2020*; *Satyshur et al., 2021*). The Minnesota Bee Atlas specimens added six additional species to the UMN Insect Collection, Minnesota's statewide repository.

The 14 stem bundles produced a total of 382 specimens, including 31 solitary nest-building wasps, 10 parasitic wasps, and 336 bees. There were 13 species of bees, including one cleptoparasitic species. The bundles of *Phragmites* stems sent by the volunteer in Brown County contained nests of *Heriades carinata, Megachile campanulae, Megachile brevis* Say, 1837, *Megachile rotundata, Megachile mendica* Cresson, 1878 and *Stelis coarctatus* Crawford, 1916. Of the bundles sent out in 2019, *Hylaeus mesillae* (Cockerell, 1896) emerged from a bundle of *Liatris ligulistylis* stems in Hennepin County. A bundle of *Asclepias incarnata* stems in St. Louis County produced *Heriades carinata, Hoplitis albifrons* (Kirby, 1837), *Hylaeus verticalis* (Cresson, 1869), *Megachile pugnata, Megachile relativa,* and *O. tersula.* Two nest-building bee species were only found in bundles: *Megachile brevis* and *Hoplitis albifron*s.

We displayed species distributions by mapping nest frequency across ecological provinces (Figs. 2, 3, Table 2). Comparison of nest frequency by province showed that total nest-building bee tunnel use per trap was similar across the LMF, EBF, and PP ($X^2 = 2.27$, $df = 2$, $p = 0.3216$) with a mean $\pm$ SE of $4.9 \pm 1.5$ in the LMF, $4.2 \pm 1.2$ in the EBF, $3.6 \pm 1.4$ in the PP (Table 3). *Osmia tersula* and *Megachile relativa* nests were significantly more frequent in the LMF than in the EBF or PP (Table 3). *Osmia lignaria* nested significantly more frequently in the LMF and EBF than in the PP. *Osmia pumila* nested significantly more frequently in the EBF than the PP and was absent from the LMF. *Heriades carinata* and *Megachile pugnata* nested significantly more frequently in the PP and EBF than the LMF. *Megachile campanulae* nested equally in PP and EBF but was absent from LMF. Nests of *Megachile inermis* Provancher, 1888, *Hylaeus annulatus* (Linnaeus, 1758) and *Hylaeus verticalis* were infrequent (present in less than 10% of nest-traps) but primarily occurred in the LMF. *Megachile rotundata, Megachile mendica, Hylaeus leptocephalus* (Morawitz, 1871), and *Hylaeus mesillae* nests were infrequent, but were primarily found in the southern half of the state across both the PP and EBF. *Megachile centuncularis* (Linnaeus, 1758) and *Heriades variolosa* (Cresson, 1872) were also infrequent but found mostly in the PP. The TAP had very few nest-traps, with only one or two nests for the species that were found there (*O. lignaria, O. tersula, Megachile relativa, Megachile pugnata,* and *Heriades carinata*). The distributions of the cleptoparasitic bees *Coelioxys moesta* Cresson, 1864, *Coelioxys alternata* Say, 1837, *Coelioxys modesta* Smith, 1854 and *S. coarctatus* tracked, to a smaller extent, those of their hosts, *Megachile relativa, Megachile pugnata, Megachile campanulae,* and *Heriades carinata*, respectively.

Tunnel-nesting bee abundance and land use were significantly correlated for the first two RDA axes according to the permutation test. Axes RDA1 (eigenvalue $= 0.05$, $F = 14.69$, $p < 0.001$) and RDA2 (eigenvalue $= 0.02$, $F = 4.99$, $p < 0.001$) of the redundancy analysis explained a cumulative 97% of the variation (Fig. 4). RDA1 primarily distinguished between grasslands and forest covers and RDA2 primarily distinguished between developed and

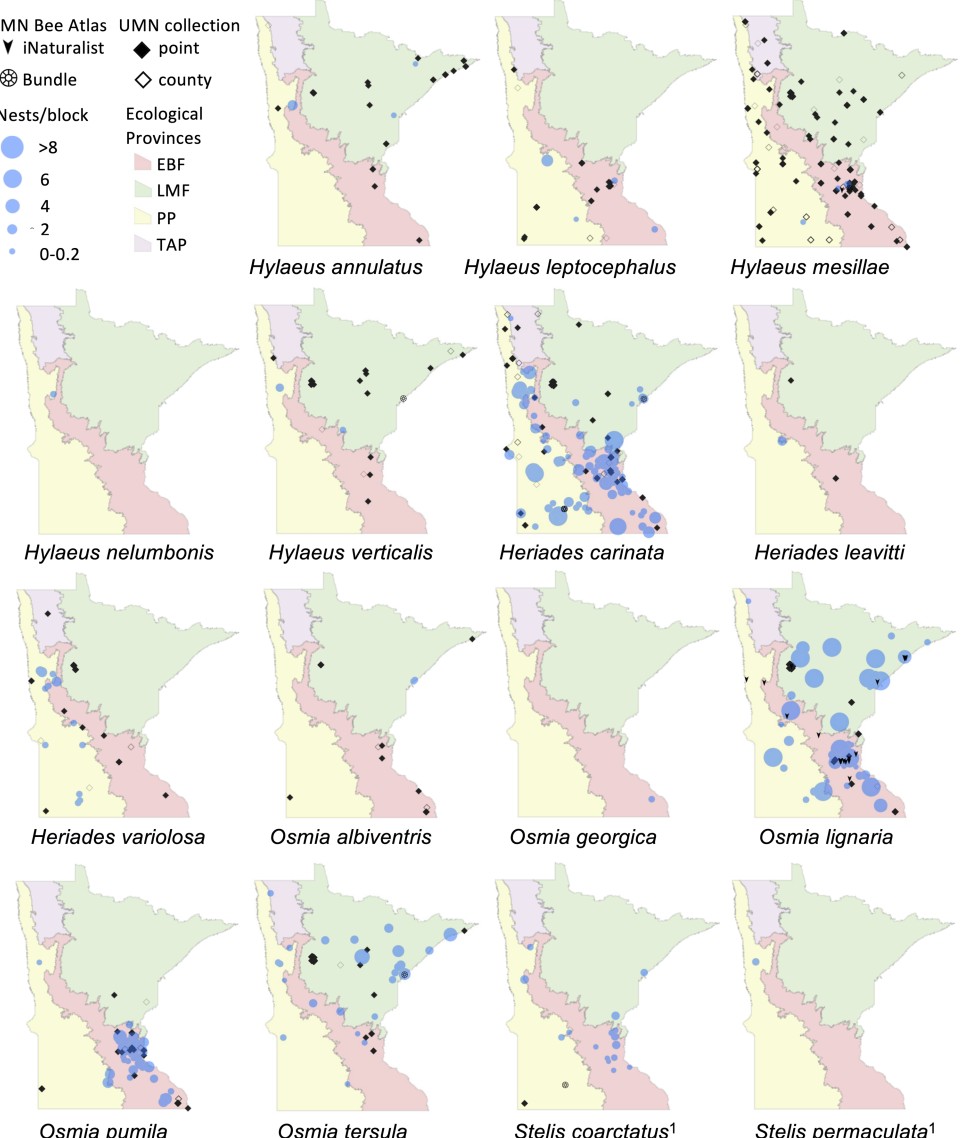

**Figure 2** **Species distribution maps of tunnel nesting bees in the genera *Heriades, Hylaeus, Osmia* and *Stelis* found from the Minnesota Bee Atlas nest traps** Data from nest traps and bundles (2016–2019) are shown as bee nests per trap, with traps grouped within 1 km locations and accounting for different numbers of traps per location. For clarity, trap locations with no nests of a species are not shown. Additional locations depicted are research-grade iNaturalist observations through October 2020 and specimens from a 2019 version of the UMN Insect Collection database, overlaid over Minnesota's four major ecological provinces. If UMN Insect Collection specimens did not have associated latitude and longitude, we used the location description to estimate the most accurate position possible. We chose the approximate center of geographic areas such as cities and state parks. If only county location was available, we placed the specimen in the county center and identified the records as such. Locations of cleptoparasitic bees are nests of their hosts from which they emerged. 1, Cleptoparasite on *Heriades*.

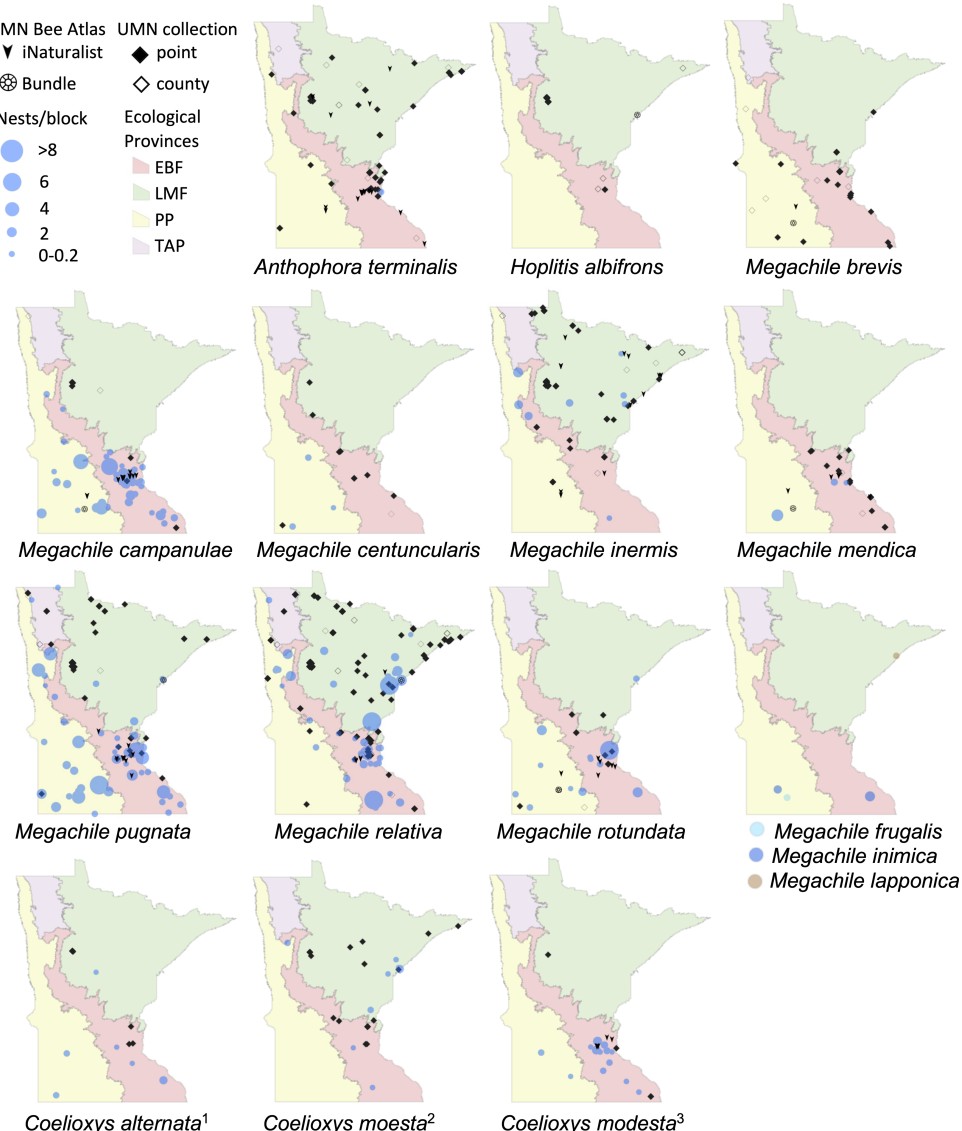

**Figure 3** Species distribution maps of tunnel nesting bees in the genera *Anthophora, Hoplitis, Megachile* and *Coelioxys,* found from the Minnesota Bee Atlas nest traps. Nest traps and bundle data (2016–2019), shown as bee nests per trap and grouped as in Fig. 2. Also shown are research-grade iNaturalist observations through October 2020 and specimens from a 2019 version of the UMN Insect Collection database, overlaid over Minnesota's four major ecological provinces. UMN Insect Collection specimens were assigned locations as in Fig. 2. Locations of cleptoparasitic bees are nests of their hosts from which they emerged. 1, Cleptoparasite on *M. pugnata*. 2, Cleptoparasite on *M. relativa*. 3, Cleptoparasite on *M. campanulae*.

grasslands (Table 4). *Heriades carinata* and *Megachile pugnata* were associated with grassland land cover (Fig. 4). *Megachile campanulae* was associated with developed land cover. *Osmia lignaria* was associated with forested land cover.

Nest phenology data from 1,041 bee nest tunnels representing 17 species was of sufficient quality to include in a summary (Fig. 5). *Osmia* completed nests earliest, with *O. lignaria*

**Table 3** **Results of linear mixed effects models of the influence of ecological provinces on frequency of tunnel-nests.** Species presented are a subset of all species collected representing those collected from more than 10% of nest blocks, representing species in the genera *Heriades*, *Osmia*, and *Megachile*. Significant results are indicated in bold. Means and standard errors are calculated from the raw data. *Post hoc* tests are the results of estimated marginal means comparisons.

| species | Mean nest frequency per block +/- s.e. | | | $X^2$ | df | *p*-value | *Post hoc* tests | |
| --- | --- | --- | --- | --- | --- | --- | --- | --- |
| | EBF | PP | LMF | | | | direction | *p*-value |
| *H. carinata* | $1.02 \pm 0.12$ | $1.34 \pm 0.21$ | $0.52 \pm 0.20$ | 6.05 | 2 | **<0.05** | EBF = PP | 0.2439 |
| | | | | | | | EBF > LMF | **0.0352** |
| | | | | | | | PP > LMF | **0.0152** |
| *O. lignaria* | $1.13 \pm 0.23$ | $0.5 \pm 0.25$ | $2.9 \pm 0.70$ | 9.22 | 2 | **<0.01** | EBF >PP | **0.0447** |
| | | | | | | | EBF = LMF | 0.1113 |
| | | | | | | | LMF > PP | **0.0027** |
| *O. pumila* | $0.79 \pm 0.12$ | $0.01 \pm 0.01$ | NA | 6.03 | 1 | **<0.01** | EBF > PP | **0.0001** |
| *O. tersula* | $0.04 \pm 0.02$ | $0.06 \pm 0.03$ | $0.90 \pm 0.20$ | 52.84 | 2 | **<0.01** | EBF = PP | 0.7153 |
| | | | | | | | LMF > EBF | **<.0001** |
| | | | | | | | LMF > PP | **<.0001** |
| *M. campanulae* | $0.42 \pm 0.05$ | $0.38 \pm 0.12$ | NA | 0.40 | 2 | 1 | EBF = PP | 0.53 |
| *M. pugnata* | $0.37 \pm 0.08$ | $0.72 \pm 0.19$ | $0.12 \pm 0.06$ | 8.66 | 2 | **<0.05** | EBF = PP | 0.1205 |
| | | | | | | | EBF > LMF | **0.0475** |
| | | | | | | | PP > LMF | **0.0043** |
| *M. relativa* | $0.27 \pm 0.09$ | $0.13 \pm 0.05$ | $0.91 \pm 0.30$ | **9.26** | **2** | **<0.001** | EBF = PP | 0.3295 |
| | | | | | | | LMF > EBF | **0.0126** |
| | | | | | | | LMF > PP | **0.0047** |
| Overall nesting | $4.2 \pm 1.2$ | $3.6 \pm 1.4$ | $4.9 \pm 1.5$ | 2.27 | 2 | 0.3216 | EBF = PP | 0.3008 |
| | | | | | | | EBF = LMF | 0.4346 |
| | | | | | | | LMF = PP | 0.1380 |

Notes.

EBF, Eastern Broadleaf Forest; PP, Prairie Parklands; LMF, Laurentian Mixed Forest.

in May, followed by *O. pumila* and then *O. tersula* near the end of June. *Osmia georgica* had only one nest, which was completed between the middle of May and the end of June. *Megachile* nests were primarily completed between 15 June and 15 August, with most *Megachile campanulae*, *Megachile pugnata*, and *Megachile relativa* completing nests near mid-July, most *Megachile inermis* and *Megachile rotundata* completing nests in late July, and most *Megachile mendica* completing nests near mid-August. We reared *Megachile relativa* from nest-traps that were brought into the lab during mid-summer, showing this species can have two generations per year in Minnesota and may have two nesting phenology peaks. *Megachile centuncularis* and *Megachile frugalis* are represented by only one nest each in late July to August. For *Megachile inimica* and *Megachile lapponica*, we have a last empty date but no full plug date, which only indicates nests were completed after about July 7 and 18 respectively. *Heriades* species primarily completed nests between 23 June and 15 August, with *Heriades carinata* slightly earlier than *Heriades variolosa* and *Heriades leavitti* Crawford, 1913.

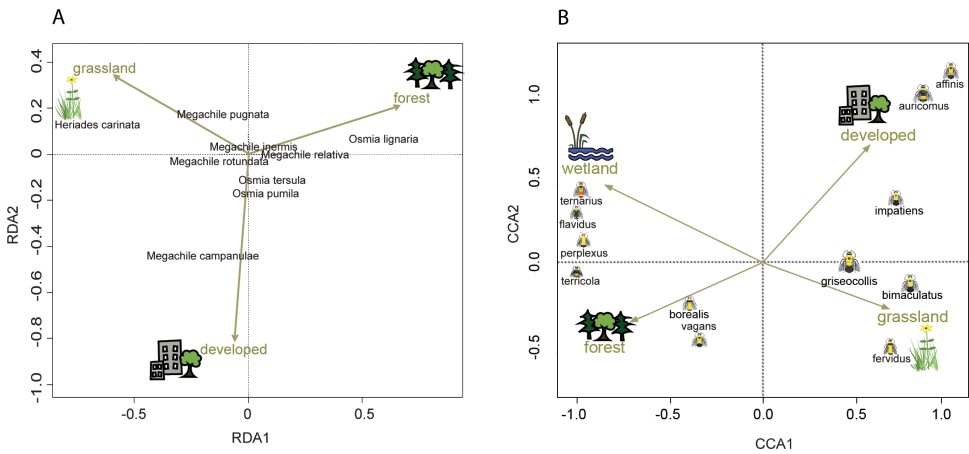

**Figure 4** **Ordination showing the relationship of land cover to tunnel-nesting bee presence and bumble bee abundance.** The location of each point relative to the arrows indicates the land cover variable associated with that species (*Palmer, 1993*). Arrow length indicates the importance of the habitat variable in predicting the variability in the model (*ter Braak, 1986*). Arrow direction indicates the strength of correlation with the axes with a small angle between arrow and axis indicating high correlation. (A) Redundancy analysis (RDA) axes 1 and 2 show the relationship of tunnel-nesting bees to land cover within 250 m of nest trap locations. (B) Constrained correspondence analysis (CCA) axes 1 and 2 show the relationship of bumble bee species to land cover within 2 km of survey locations. Axis 1 eigenvalue $=0.60$, $F = 66.32$, $p < 0.001$, axis 2 eigenvalue $=0.10$, $F = 9.62$, $p < 0.001$.

**Table 4** **Biplot scores for constraining variables of land cover related to presence of tunnelnesting bee species or bumble bee species abundance.** The forest category combines deciduous, mixed, and evergreen forest. All levels of development were combined into the category. The grassland category includes grasslands/herbaceous and pasture/hay. The wetland category includes woody wetlands and emergent herbaceous wetlands. Correlations with absolute values $\geq 0.5$ are bolded.

| Tunnel-nesting bees | RDA1 | RDA2 | RDA3 | |
|---|---|---|---|---|
| Developed | −0.07 | **−.99** | −0.02 | |
| Forest | **0.82** | 0.25 | **−0.51** | |
| Grassland | **−0.73** | 0.42 | **−0.53** | |
| **Bumble bees** | **CCA1** | **CCA2** | **CCA3** | **CCA4** |
| Developed | **0.57** | **0.62** | 0.41 | 0.34 |
| Wetland | **−0.83** | 0.41 | −0.18 | −0.30 |
| Forest | **−0.76** | 0.35 | **0.50** | −0.11 |
| Grassland | **0.67** | −0.24 | 0.03 | **−0.71** |

## Bumble bees

Volunteers recorded 9,186 individuals belonging to 17 bumble bee species during 1,330 10-minute observations at survey stops. Volunteers observed zero bumble bees at 220 out of 1,330 survey stops. Volunteers observed no bees across all five survey stops along a route for 10 route runs, representing seven different routes. Several species of conservation concern were documented, including 17 *B. affinis* along four routes, 103 *B. terricola* along 14 routes, and 22 *B. pensylvanicus* along 11 routes (Table 5). Patterns of abundance from survey routes added information on regional prevalence of bumble bee species in comparison to historic

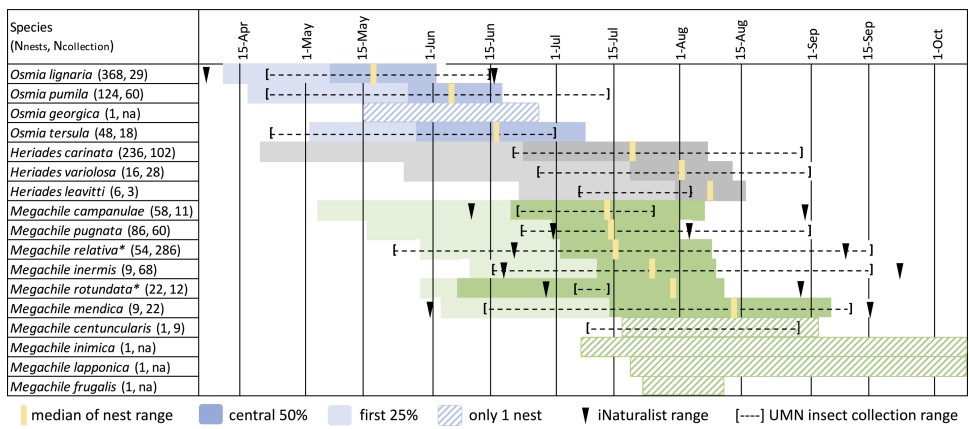

**Figure 5 Phenology of tunnel-nesting bee nest completion.** We calculated nest completion date ranges, equal to the last empty tunnel date until the first full plug date, for all nests with observation quality rated "medium" or "high". Each day in the nest completion date range was assigned equal probability. These probabilities were summed over all nests with sufficient quality observations and the median value was determined, indicating the date where nests were equally likely to be completed before or after. We also calculated the 0.25 and 0.75 quartile values, which bound the central 50% when most nests were likely completed. Because bees may be active for several weeks before nests are completed and plugged, we want to emphasize the beginning of the period and indicate the earliest 25% of ranges with light shading. The genus *Osmia* is shaded in blue, *Heriades* in gray, *Megachile* in green. Each species name is followed by parenthesis in which we list the number of nests used to calculate phenology from Minnesota Bee Atlas nest traps from 2016–2018, then the number of UMN insect collections specimens. An asterisk (*) indicates species with more than one generation per year in Minnesota.

and biodiversity portal records that did not include survey effort (Figs. 6, 7). For example, while *B. rufocinctus* was present in records from all four ecoregions, surveys showed that *B. rufocinctus* was most abundant in the EBF. The composition and total bumble bee abundance varied among ecological provinces (Table 5). The most common bumble bees in the EBF were *B. impatiens* (1,781), *B. bimaculatus* (1,109), *B. vagans* group (756), and *B. griseocollis* (733). The most common bumble bees in the PP were *B. griseocollis* (102), *B. bimaculatus* (77), and *B. impatiens* (55). *Bombus ternarius* (1,466) and *B. vagans* group (1,116) were the most common bumble bees in the LMF. Total bumble bee abundance within a route in a year differed among ecological provinces ($X^2 = 12.03$, $df = 2,78$, $p < 0.01$) with bee abundance per route lower in the PP than the EBF or the LMF (Fig. 8, Table 6).

Bumble bee species abundance and land use were significantly correlated for the first three canonical axes according to the Monte Carlo permutation test. Bumble bee species Axes CCA1 (eigenvalue = 0.60, $F = 66.32$, $p < 0.001$) and CCA2 (eigenvalue = 0.09, $F = 9.62$, $p < 0.001$) of the correspondence analysis explained a cumulative 42% of the variation (Fig. 4). CCA1 primarily distinguished between grasslands and wetlands covers and CCA2 primarily distinguished between developed and grassland covers (Table 4). Habitat associations for species with lower abundances may be due to chance (*Legendre & Legendre, 2012*), leading to caution interpreting habitat associations for these species due to their low abundances: *B. affinis* (17), and *Bombus flavidus* Eversmann, 1852 (36). *Bombus*

**Table 5 Bumble bee species total abundance and abundance within three ecological provinces.** Species are ordered from greatest to least total abundance.

| *Bombus* species | Total | EBF | LMF | PP |
|---|---|---|---|---|
| *ternarius* (Say, 1873) | 2,069 | 602 | 1,466 | 1 |
| *impatiens* (Cresson, 1863) | 1,975 | 1,781 | 140 | 54 |
| *vagans* group[a] | 1,904 | 756 | 1,116 | 32 |
| *bimaculatus* (Cresson, 1863) | 1,257 | 1,109 | 71 | 77 |
| *griseocollis* (DeGeer, 1773) | 977 | 733 | 142 | 102 |
| *borealis* (Kirby, 1837) | 252 | 68 | 173 | 11 |
| *auricomus* (Robertson, 1903) | 145 | 116 | 7 | 22 |
| *rufocinctus* (Cresson, 1863) | 143 | 122 | 21 | 0 |
| *fervidus*[b] (Fabricius, 1798) | 131 | 103 | 14 | 14 |
| *terricola*[b] (Kirby, 1837) | 103 | 34 | 69 | 0 |
| *perplexus* (Cresson, 1863) | 71 | 28 | 43 | 0 |
| *citrinus* (Smith, 1854) | 42 | 20 | 20 | 2 |
| *flavidus* (Eversmann, 1892) | 36 | 20 | 16 | 0 |
| *pensylvanicus*[b] (DeGeer, 1773) | 22 | 20 | 0 | 2 |
| *affinis*[b] (Cresson, 1863) | 18 | 17 | 1 | 0 |
| *insularis* (Smith, 1861) | 2 | 1 | 1 | 0 |

**Notes.**
[a]*Bombus vagans* group includes *B. vagans* (Smith, 1854) and *B. sandersoni* (Franklin, 1913).
[b]Categorized with IUCN status vulnerable or critically endangered (Hatfield, 2015).
EBF, Eastern Broadleaf Forest; LMF, Laurentian Mixed Forest; PP, Prairie Parklands.

*fervidus*, *B. griseocollis*, and *B. bimaculatus* were associated with grassland land cover (Fig. 4). *Bombus vagans* group, *B. borealis,* and *B. terricola* were associated with forested land cover. *Bombus ternarius*, *Bombus perplexus* Cresson, 1863, and possibly *B. flavidus* were associated with forested and wetlands land covers. *Bombus impatiens*, *Bombus auricomus* (Robertson, 1903), and possibly *B. affinis* were associated with developed land cover. An alternative CCA using presence-absence data instead of abundance data for bumble bees is available in Appendix S4 to address possible aggregation effects from nest proximity.

## DISCUSSION

The Minnesota Bee Atlas project was made possible by the contributions of over 2,500 project volunteers and other iNaturalist users across three sampling protocols who recorded 30%, or 151, of the approximately 500 bee species known in Minnesota (*Portman et al., 2023*). Each sampling protocol contributed different and complementary data, indicating that multiple sampling levels would be useful in future bee monitoring projects. Through iNaturalist, volunteers reported new locations for *B. affinis*, as well as recording several other rare bumble bees and the first state record of an adventive species. Nest-traps in this project produced baseline range data for 31 species, including four new state records, and expanded the known range for 16 of those species. We also found ecological province associations for six tunnel-nesting species and landcover associations for four species. Volunteer-collected data provided relative nesting seasonality of bee species and indicated some species with multiple generations per year. Bumble bee surveys examined

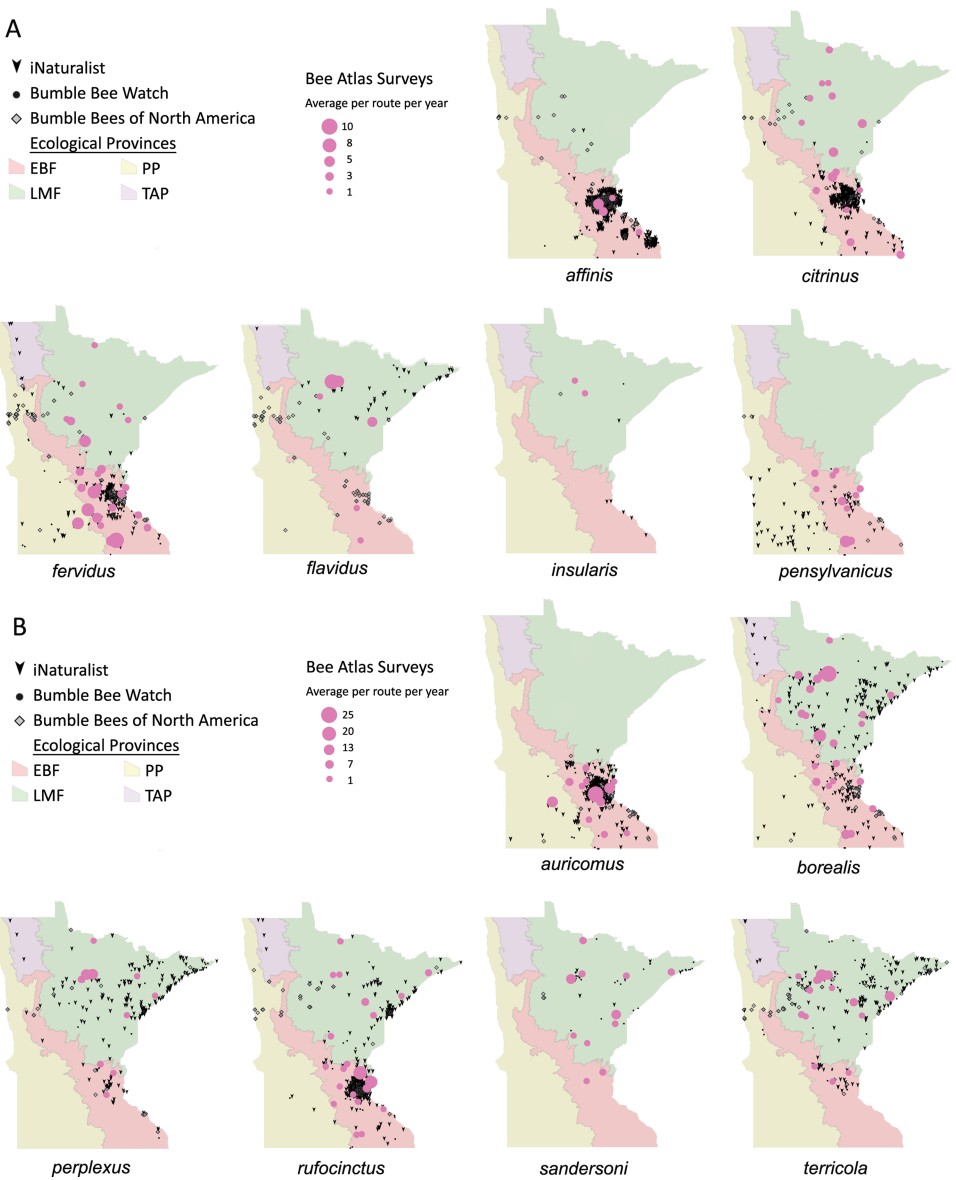

**Figure 6** **Species distribution maps for bumble bee species found during Minnesota Bee Atlas surveys with maximum average abundances between 1 and 25 bees per route per year.** The Atlas observations are overlaid over Minnesota's four major ecological provinces. We summarized survey information as the total abundance per species per route per year and displayed the average abundance per route per year for routes that were sampled over multiple years. (A) Species with maximum abundances of 10 or fewer. (B) Species with maximum abundances between 11 and 25. Additional records displayed are from iNaturalist from 2014 to 2020, Bumble Bee Watch from 2010 to 2022, and specimen-based Minnesota records from the Bumble Bees of North America database from 1889 to 2020 (*Richardson, 2021*). An asterisk (*) indicates that species abundances for *B. sandersoni* are likely lower due to exclusion of records that could not be distinguished between *B. vagans* and *B. sandersoni*.

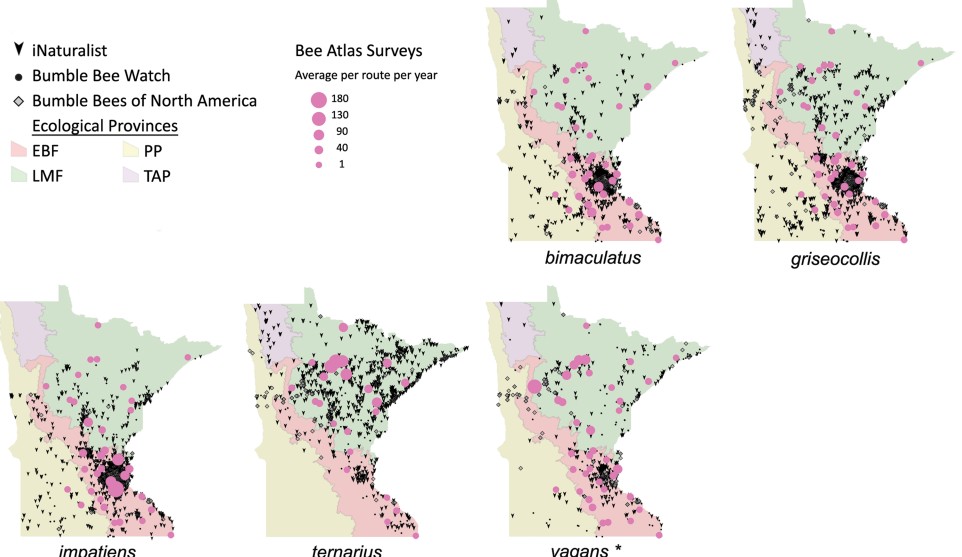

**Figure 7** **Species distribution maps for bumble bees found during Minnesota Bee Atlas surveys with maximum average abundances per route per year between 25 and 180.** These observations are overlaid over Minnesota's four major ecological provinces. Additional records displayed are from iNaturalist from 2014 to 2020, Bumble Bee Watch from 2010 to 2022, and specimen-based Minnesota records from the Bumble Bees of North America database from 1889 to 2020 (*Richardson, 2021*). An asterisk (*) indcates that species abundances for *B. vagans* are likely lower than expected due to exclusion of records that could not be distinguished between *B. vagans* and *B. sandersoni*.

abundances across ecological provinces, indicating potential benefit of a regional focus on bumble bee habitat management, as well as possible habitat associations for species of conservation concern. The ecological associations and patterns of abundance discovered by the Minnesota Bee Atlas can inform management decisions to improve pollinator conservation actions and recovery of endangered species.

## iNaturalist

There are strengths and limitations to using iNaturalist to study bees. One clear strength is the large number of observers, which increases the chances of finding rare species (*Donnelly et al., 2014*; *Wilson et al., 2020*), especially bumble bees, which were most frequently photographed and identified in our project. Many bumble bee species are becoming less abundant and experiencing reductions in their geographic ranges, making information about their status particularly important for conservation efforts (*Goulson, Lye & Darvill, 2008*; *Hatfield et al., 2015*; *Beckham & Atkinson, 2017*). New location information for *B. affinis* is important for recovery plans for this endangered bee (*US Fish and Wildlife Service, 2021*). The iNaturalist records of *B. frigidus* and *B. nevadensis*, which were not found in the more structured surveys, also illustrate the utility of the large number of observers and widespread observations on the platform.

A second strength of iNaturalist is that observations are rapidly available, making the platform useful for monitoring adventive species that can be quickly identified to research

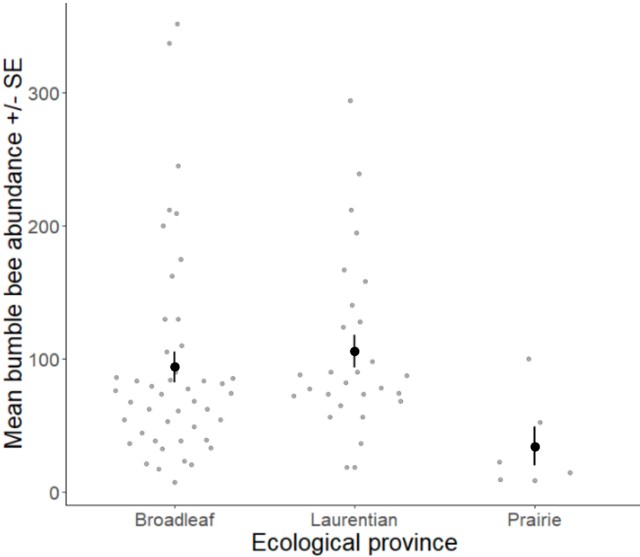

**Figure 8 Bumble bee abundance across the Eastern Broadleaf Forest, Laurentian Mixed Forest, and Prairie Parkland ecological provinces.** Bumble bee abundance is shown as the average abundance per route per year for routes with three completed survey dates within a year. A single route from the TAP ecological province was combined with routes from the PP ecological province due to the low sample size in this province and ecological similarity.

**Table 6 Results of linear mixed effects model of influence of ecological provinces on overall bumble bee abundance.** Bee abundances are log-transformed. Significant results are indicated in bold. *Post hoc* tests are the results of estimated marginal means comparisons.

| Fixed effects | $X^2$ | df | *p*-value | | $\beta$ +/- 95% CI | *Post hoc* tests direction | *p*-value |
|---|---|---|---|---|---|---|---|
| Ecological province | 12.03 | 2 | **<0.01** | EBF | 4.28 (3.75–4.69) | EBF = LMF | 0.88 |
| | | | | LMF | 4.40 (4.02–4.79) | EBF >PP | **<0.01** |
| | | | | PP | 3.02 (2.29–3.74) | LMF >PP | **<0.01** |
| Year | 3.26 | 4 | 0.52 | 2016 | 3.84 (3.37–4.31) | | |
| | | | | 2017 | 3.83 (3.47–4.19) | | |
| | | | | 2018 | 3.81 (3.45–4.17) | | |
| | | | | 2019 | 4.04 (3.69–4.39) | | |
| | | | | 2020 | 3.99 (3.64–4.33) | | |
| Random effects | | | Variance +/- SD | | | | |
| Route | | | 0.40 +/- 0.63 | | | | |
| Residual | | | 0.17 +/- 0.41 | | | | |

**Notes.**

EBF, Eastern Broadleaf Forest; PP, Prairie Parklands; LMF, Laurentian Mixed Forest.

grade. Previously documented in neighboring states (*Parys, Tripodi & Sampson, 2015*), *Megachile sculpturalis*, an introduced species with an expanding range, was recorded for the first time in Minnesota in the first year of the Bee Atlas project. Although it was only recorded once in the Minnesota Bee Atlas iNaturalist project, it is a large and easily recognized bee, and opportunistic participatory science platforms have been important to monitoring its spread in Europe (*Le Féon et al., 2018*; *Flaminio et al., 2021*; *Dubaić et al.,*

Peer] _________________________________________________

*2022*). The fact that *Megachile sculpturalis* has only been recorded once in the five years of the project may indicate that it is reaching either the northern or western limits of its range in North America, or it could indicate the low population densities typical of the early stages of colonization (*Dubaić et al., 2022*). Increased monitoring effort is needed to assess its status and potential impact. With outreach to engage public interest, the Minnesota Bee Atlas iNaturalist project may be able to produce accurate and up to date distribution maps for *Megachile sculpturalis*, allowing biologists to determine its spread in the state.

One limitation of iNaturalist is that observations do not reflect relative abundance. Larger bees comprise the majority of observations, both non-research and research-grade, with over half of non-research grade observations from the families Apidae and Megachilidae. Among the larger bees, a subset of more easily identified bees, bumble bees and honey bees, make up 85% of research-grade observations. This is consistent with other opportunistic participatory science programs, which either focus on bumble bees exclusively or broad bee groupings (*Beckham & Atkinson, 2017*; *Maher, Manco & Ings, 2019*; *Flaminio et al., 2021*; *Griffin et al., 2021*). In strong contrast, sweep netting collections in this region show high abundances of bees from the family Halictidae (*Lane et al., 2020*; *Evans et al., 2022*). For many other bee groups, and especially smaller species, existing identification methods require expert examination of physical specimens to assign species-level identifications (*Le Féon et al., 2016*; *Woodard et al., 2020*; *Flaminio et al., 2021*). If iNaturalist records are used to describe the structure of the bee community, one should keep in mind that some groups of bees are likely to be overlooked because of their small size, nondescript coloring, or habitat specialization. However, the likelihood of identification may be improved with training to improve photo quality and advancements in artificial intelligence.

## Tunnel-nesting bees

Nest traps and stem bundles combined with iNaturalist observations enhanced our understanding of species distributions in Minnesota for 32 tunnel-nesting species. For 16 species, our project expanded the known geographic extent of their distribution in the state compared to the UMN Insect Collection. We documented that the ranges of five cleptoparasitic bee species mimicked that of their hosts but with a smaller geographic spread. This may indicate the range in which the host bees have a large enough population to support these parasitic bees (*Sheffield et al., 2013*). The collection of four new species records for the state along with rarely collected species is consistent with *Westphal et al. (2008)*, who found numerous species in nest-traps in Europe that were not recorded with any other sampling methods. It may also reflect our expansion of collection efforts over the whole state or possible recent changes in species' ranges.

Clarifying distributions allows us to start associating bees with climates and habitats, as well as providing baseline data for future comparisons. By using standardized, repeatable methods to survey the whole state simultaneously, we were able to compare nest frequency and explore ecological province and landcover associations. Province associations could be due to climatic or plant community differences. For example, both factors may influence the distribution of *Megachile relativa*. This species can have lower supercooling points than *Megachile rotundata*, which allowed *Megachile relativa* to survive winter outdoors in

Alberta, Canada (*Krunic & Salt, 1971*) and may contribute to its northern distribution and association with the LMF in this study. The LMF plant community could also contribute to this observed association. The LMF is characterized by broad areas of conifer forest, mixed hardwood and conifer forests, and conifer bogs and swamps (*Hanson & Hargrave, 1996*). Despite our finding of no association of *Megachile relativa* with forested land cover, previous observations showed that this species preferred nest sites at woodland edges in Wisconsin (*Medler & Koerber, 1958*). Other bee species showed associations that were counter to our expectations based on current knowledge of their biology. We expected the bee species that use resin for nest construction, *Heriades carinata* and *Megachile campanulae*, to nest more frequently in the LMF due to the dominance of many resin-producing trees in the LMF, including *Pinus, Abies, Picea*, and *Populus* spp. (*Minnesota Department of Natural Resources, 2022*) and accounts of conifer resin use in *Heriades* and *Megachile campanulae* (*Medler & Lussenhop, 1968*; *Maciel de Almeida Correia, 1977*; *Macivor & Salehi, 2014*). However, we found that *Megachile campanulae* was absent from the LMF and that *Heriades carinata* nested more frequently in the PP and EBF than the LMF and was associated with grassland land cover. The plant communities of the PP and EBF may contain acceptable resin sources for these bees. Alternatively, the availability of resin plants as a nesting resource may not limit distribution. *Westerfelt, Weslien & Widenfalk (2018)* found that tunnel nesting bee nest abundance could be predicted by both nest substrate and food plant availability, but to different degrees for pollen generalists and specialists. *Heriades carinata* is considered polylectic, but *Megachile campanulae* has been associated with flowers in the genus *Campanula,* and their distribution may be predicted more by food resources. This bee was associated with developed land cover in our study, and high abundance of the weedy plant *Campanula rapunculoides* in developed areas could be a driver in their nesting success. In another case, the province associations we found differ from past records. We found significantly higher nest frequency for *Megachile pugnata* in the EBF and PP, while specimens in the UMN Insect Collection were predominantly in the EBF and LMF. This discrepancy could be due to different collecting efforts or could reflect previous landscapes or distributions (*Gardner & Spivak, 2014*). We found an association of *Megachile pugnata* with grassland land cover, which could explain their higher frequency in the PP and the EBF.

Five of the nine tunnel-nesting bee species tested in this study showed no association in the land cover analysis. This may indicate that a single broad land cover category does not capture the habitat elements to which many tunnel-nesting bees are responsive. In addition, it should be noted that the distributions of *O. lignaria* and *Megachile rotundata* may be influenced by human management, including commercial sales, in addition to climatic differences and plant communities.

Although nest-traps have been shown to be a reliable way to assess ecological association of tunnel-nesting bee species (*Staab et al., 2018*), nest-traps typically only sample a portion of the tunnel-nesting bee community (*Westphal et al., 2008*; *Prendergast et al., 2020*). Several factors may have contributed to the non-detection of tunnel-nesting bee species in this study, which should not be interpreted as absence. It is possible that species may utilize nest-traps less frequently in areas with more suitable natural nesting substrates (*Westphal et*

al., 2008; *Carper & Bowers, 2017*), which is a complicating factor for this sampling method. However, in this study, overall bee nest frequency was statistically similar across all ecological provinces, forested or otherwise (Table 3). Some nests produced no identifiable offspring due to parasitism or other causes. These nests were left out of all analyses. As we saw in this study, some tunnel-nesting bee species in Minnesota may have more than one generation per year. Species emerging before our fall nest trap collection would not be captured if they did not re-nest in the traps. Rare species take more effort to detect, and even with our full coverage of the state, three years of sampling, and focus on natural habitats, we may have sampled too small a proportion of bees to reliably find some rare species, or species that prefer rare habitats. Solid wood traps may not be an acceptable or preferred nest substrate for some tunnel-nesting bee species. Although *Osmia* and *Megachile* are often considered tunnel-nesting genera, a proportion of species in both genera nest in the ground, and we would not have expected them in this study (*Cane, Griswold & Parker, 2007*; *Sheffield et al., 2011*; *Rightmyer, Griswold & Brady, 2013*). Similarly, bees in the genus *Ceratina* Latreille are obligate stem excavators and would not be expected (*Rehan & Richards, 2010*; *Vickruck et al., 2011*). Two species that we collected rarely in the Bee Atlas, *Hylaeus mesillae* and *Anthophora terminalis* (Cresson, 1869), were common in UMN Insect Collection records, suggesting that wood block nest-traps are a less effective sampling method for these species. *Anthophora terminalis* is known from fallen or rotting wood substrates (*Cockerell, 1903*; *Sladen, 1919*; *Medler, 1964*), as are *Megachile frigida* Smith, 1853 and *Osmia bucephela* Cresson, 1864 (*Stephen, 1956*; *Krombein, 1967*) which we did not collect. Pithy or hollow stems of many plant species are also used as nest substrates (*Satyshur & Evans, 2021*) and might be preferred by some bees. Our stem bundles did not produce enough bee nests to distinguish any preference between plant stem species but did produce two bee species not collected in our wood nest-traps: *Megachile brevis* and *Hoplitis albifrons*. *Hoplitis* species and *Hylaeus messillae* are frequently found in stems (*Parker & Bohart, 1966*; *Medler & Lussenhop, 1968*) but were rare in this study. *Megachile brevis* is known from a wide variety of substrates including dead stems, ground, leaves and under dried cattle dung (*Michener, 1953*). Some Minnesota species not found in this study, such as *Megachile montivaga* Cresson, 1878 (*Orr, Portman & Griswold, 2015*) and *Osmia atriventris* Cresson, 1864 (*Fye, 1965*) are also known from stems. Future studies of tunnel-nesting bees are likely to sample a larger proportion of the community by using both wood and stem substrates. A more targeted study, returning to known collection areas and looking for species that have not been recorded in Minnesota in recent years is warranted.

In addition to distribution data, we collected data on nesting phenology, which returned a date range when tunnel nesting bee species are likely to complete nesting and indicated the relative seasonality of species. Volunteer observations also allowed us to catch *Megachile relativa* emerging both mid-summer and the following spring. This agrees with the bivoltine life cycle for *Megachile relativa* found in Wisconsin (*Medler & Koerber, 1958*) and expands the known range of bivoltinism into Minnesota. It is important to remember that the phenology event volunteers recorded was nest plugs, which are made after a nest is completed. Therefore, the bee's active period likely begins several weeks earlier. Despite

this, in 10 of the 17 species we have data for, nest plugs were observed several days to several weeks earlier than the range of collection dates for the same species in the UMN Insect Collection (Fig. 5). This could be due to the large increase in records and full season of data collection made possible by participatory science (*Soroye, Ahmed & Kerr, 2018*; *Dubaić et al., 2022*). Another possibility is that earlier recorded activity periods are the result of advancing phenology with climate change. *Bartomeus et al. (2011)* compared collection dates of museum specimens collected between 1880 and 2010 for 10 bee species in northeastern North America, including two of the species in this project. They found an average phenological advance of 10.4 days. The phenological data we have recorded helps define these bees' temporal habitats and lays the groundwork for assessing changes.

### Bumble bees

The bumble bee surveys of the Minnesota Bee Atlas project used consistent survey effort across routes, providing the opportunity to examine patterns of bumble bee abundances and species associations with land use, all of which have been difficult to do from museum collections or biodiversity portal observations alone. We have reliable information on ranges of Minnesota bumble bees due to numerous records of bumble bee species courtesy of the Bumble Bees of North America database (*Richardson, 2021*). Our surveys not only confirm ranges, such as the northern distributions of *B. ternarius*, *B. terricola*, *B. borealis*, *B. flavidus*, and *B. perplexus*, but also provide insight into bumble bee community structure. For example, although *B. griseocollis* is present throughout the state, they are the dominant bumble bee community members in only two of the three examined ecological provinces (PP and EBF). Further exploration could reveal specific ecological drivers of this pattern. Although we identified many of the submitted photographs for *B. vagans* and *B. sandersoni* to species level to create maps showing their distributions, *B. vagans* had to be combined with *B. sandersoni* for comparisons of abundance and habitat associations, because many observations could not be distinguished. Future volunteer surveys may be able to distinguish these species as the quality of cameras available to volunteers increases. Minnesota bumble bees not found on survey routes include *B. frigidus*, *Bombus huntii* Greene 1860, *Bombus variabilis* (Cresson, 1872), *Bombus ashtoni* (Cresson, 1864) (sometimes considered to be conspecific with *Bombus bohemicus* Seidl, 1837), *Bombus fraternus* (Smith, 1854), and *B. nevadensis*. This is likely because these species are extremely rare, their ranges barely extend into Minnesota, or because they are not usually found on roadsides.

Bumble bee abundance information gathered by the bumble bee surveys provides important baseline information and informs management decisions to support bumble bees. Many studies of bumble bee decline rely on relative rather than absolute abundances of bumble bees (*Colla & Packer, 2008*; *Koch, 2011*; *Cameron et al., 2011*). While this approach helps us understand shifts in communities, it does not answer questions about broad trends in abundance, a key conservation concern. Even with consistent survey effort, we do not expect counts of bumble bees on flowers to reflect true population sizes at a particular site due to possible aggregation effects from floral abundance or nest proximity (*Harder, 1986*; *Hines & Hendrix, 2005*; *Geib, Strange & Galen, 2015*). However, we do expect to get metrics that can be repeated across a broad geographical and temporal scale to detect

bumble bee abundance patterns. The observed lower bumble bee abundance in the PP could indicate lower bumble bee abundance in that ecological province overall, could indicate differences in the attractiveness of roadside habitat to foraging bumble bees between ecological provinces due to concentration or dilution effects with varying floral abundance in non-roadside habitats, or could be an artifact of the smaller number of routes that were run in this ecological province. Our volunteers did not gather information on the floral cover at survey sites, but volunteers in the PP more frequently reported difficulty finding areas with flowers along their assigned routes. A recent study in the same area in restored prairies found abundant bumble bee populations, indicating that the PP is not depauperate of bumble bees across habitats (*Lane et al., 2020*). Since these prairie habitats make up less than 1% of the PP (*Lark et al., 2019*), increasing floral availability and abundance along the extensive amount of roadside habitat in the PP could provide support to bumble bees in these isolated prairie remains, particularly along roads with lower traffic to reduce risks from vehicle collisions and road pollution (*Keilsohn, Narango & Tallamy, 2018*; *Shephard et al., 2022*).

Association of bumble bee species with surrounding land cover can help assess habitat needs of different bumble bee species. While our survey routes were limited to roadside habitats, the predominant land uses surrounding our survey routes varied, providing an opportunity to examine the influence of land use on bumble bees. Many of the associations we found are similar to those found in an examination of land cover and the probability of bumble bee occurrence in Vermont (*Richardson et al., 2019*). We both found *B. vagans* group and *B. terricola* to be positively associated with forested land cover, *B. fervidus*, *B. griseocollis*, and *B. bimaculatus* to be positively associated with grassland land covers, and *B. impatiens* to be positively associated with developed land cover. Our study included several species not present in the Vermont survey. The positive association of *B. auricomus* and possibly *B. affinis* with developed land cover have not been previously reported to our knowledge.

Most recent records for *B. affinis* have been contributed by the public and are associated with urban areas in Minnesota, Wisconsin, Iowa, and Illinois (*US Fish and Wildlife Service, 2021*). It is not clear whether this phenomenon is due to more people in urban areas looking for rare species and contributing records to public monitoring or whether *B. affinis* is associated with developed areas. Since our survey routes were spread throughout the state across a wide range of habitats, our finding of a possible association between *B. affinis* and developed land cover indicates that the phenomenon may not be entirely due to increased participation in monitoring in urban areas. Historically, *B. affinis* nests have been noted to be associated with urban areas, and have been found near houses (*Medler, 1963*). The possible association of a federally protected endangered species with developed land has important implications for conservation strategies, which often take advantage of publicly owned land. Conservation efforts on private, multi-use property have additional complications (*Kamal, Grodzińska-Jurczak & Brown, 2015*).

## CONCLUSIONS

Through four field seasons and participation from over 2,500 members of the public, the Minnesota Bee Atlas used uniform methods to survey bees across Minnesota. Our findings include (1) documentation of rare and endangered bees of conservation concern, (2) extension of known ranges for tunnel-nesting species, (3) bee associations with ecological provinces, (4) nesting phenology data for tunnel-nesting species, (5) state-wide abundance patterns for bumble bees in roadside habitats, and (6) habitat associations for bumble bee and tunnel-nesting bee species. In addition, we documented new state records and gathered baseline, replicable data on tunnel-nesting bees and bumble bees across the state. An added benefit of our program is the increased awareness of pollinator conservation among our volunteers, who continue to contribute to other participatory science projects, submit thousands of iNaturalist records, and lead their own outreach efforts. Our findings support several habitat management recommendations. Broad-scale land use changes have occurred over the last 150 years leading to reduction of natural habitat to less than 2% across all ecological provinces due to conversion to cropland and managed forests (*Wendt & Coffin, 1988*), impacting both nesting and foraging habitats for bees (*Benton, Vickery & Wilson, 2003*; *Holzschuh, Steffan-Dewenter & Tscharntke, 2010*; *Le Féon et al., 2010*). With similar abundances of tunnel-nesting bees in the prairie and two forested ecological provinces, and with a variety of habitat associations among species, a broad range of regions and habitats are suitable targets for tunnel-nesting bee habitat enhancement. Providing a variety of stem and wood nesting substrates mimicking natural density may support nesting. Interpreting our findings from bumble bee abundance patterns, we found a need for increased floral availability in roadside habitat in the PP ecological province to support bumble bees, which could also support other pollinators.

The baseline data we provided can be compared with future surveys using comparable methods to examine trends in populations of tunnel-nesting bees and bumble bees, with the understanding that the distributions we have documented have been influenced by current land use and climate as well as historic land use changes. These comparisons can help assess the impact of subsequent pollinator conservation efforts as well as long-term stressors such as climate change. We recommend the following improvements to survey methods: (1) Publicizing information about *Megachile scupturalis* and other easily identified introduced species and engaging iNaturalist users in tracking their spread in the state, (2) using stem substrates in conjunction with wood substrates for nest-traps to increase the number of species captured, (3) targeted nest-trap surveys in regions and habitats that were underrepresented in this project, (4) the inclusion of a wider variety of habitat types in surveys to improve assessment of the bumble bee community, and (5) additional participant training to assess habitat in survey locations to help identify habitat improvements needed to support bumble bees in different regions.

Overall, the Bee Atlas project shows the strength of involving the public in scientific research to cover the geographic range of a state with methods that enable comparison of relative and absolute abundance in different habitats and to document species that have not been discovered using other methods. Coupled with professional experts, trained

volunteers provided vital information that University researchers alone would have been unable to collect, showing the value of public participation in bee research and monitoring.

## ACKNOWLEDGEMENTS

Thank you to all our volunteers who hosted nest blocks, adopted bumble bee survey routes, submitted pictures to iNaturalist, and helped with nest block construction, bee rearing and pinning in the lab. We could not have done this without you. We thank V. Scott and A. Rose of the Bees' Needs project of Colorado for initial conversations and initial plug material photos. Thank you to Jason Gibbs for confirming identifications for new state records, Ryan Oram for verifying representatives of *Hylaeus*, Michael Orr and Sam Droege for confirming identification of *Anthophora terminalis*, John Luhmen for identifying Ichneumonid specimens and providing consultation on Chalcidoidea identification, and Jorge González and Mike Gates for confirming *Kocourekia* cf. *debilis* identification. The iNaturalist identifications from Zach Portman, John Ascher, Joel Neylon, Tony Ernst, and Brian Dagley were invaluable for data quality on that platform and we thank them for their time and efforts. Thank you to Clarence Lehman for consultation and editing, and Robin Thomson for curation of the University of Minnesota Insect Collection. The feedback of five anonymous reviewers helped polish the manuscript.

### Funding

Funding for this project was provided by the Minnesota Environment and Natural Resources Trust Fund as recommended by the Legislative-Citizen Commission on Minnesota Resources (LCCMR). M.L. 2015, Chp. 76, Sec. 2, Subd. 03g as extended by M.L. 2019, First Special Session, Chp. 4, Art. 2, Sec. 2, Subd. 19 as extended by M.L. 2020, First Special Session, Chp. 4, Sec. 2. The funders had no role in study design, data collection and analysis, decision to publish, or preparation of the manuscript.

### Grant Disclosures

The following grant information was disclosed by the authors:
The Minnesota Environment and Natural Resources Trust Fund as recommended by the Legislative-Citizen Commission on Minnesota Resources (LCCMR).

### Competing Interests

The authors declare there are no competing interests.

### Author Contributions

- Colleen D. Satyshur conceived and designed the experiments, performed the experiments, analyzed the data, prepared figures and/or tables, authored or reviewed drafts of the article, and approved the final draft.
- Elaine C. Evans conceived and designed the experiments, performed the experiments, analyzed the data, prepared figures and/or tables, authored or reviewed drafts of the article, and approved the final draft.

- Britt M. Forsberg conceived and designed the experiments, performed the experiments, authored or reviewed drafts of the article, and approved the final draft.
- Thea A. Evans performed the experiments, authored or reviewed drafts of the article, and approved the final draft.
- Robert Blair conceived and designed the experiments, authored or reviewed drafts of the article, and approved the final draft.

## Field Study Permissions

The following information was supplied relating to field study approvals (i.e., approving body and any reference numbers):

Research permits were approved by the Minnesota Department of Natural Resources (2016-29, 2016-4R, 201705, 2017-9R, 201822, 2018-15R).

## Data Availability

The data and code are available at GitHub:

https://github.com/elaineceleste/Minnesota-Bee-Atlas.

## Supplemental Information

Supplemental information for this article can be found online at http://dx.doi.org/10.7717/peerj.16146#supplemental-information.

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
