# Peer review of "Determining Minnesota bee species’ distributions and phenologies with the help of participatory science"

_PeerJ, doi:10.7717/peerj.16146_

## Round 0.1 · original submission · Minor Revisions

Dear authors,

Many thanks for your submission. We have received three reviews, all of which have suggested some excellent edits. Please look through the and address them to the best of your ability. I think they will ameliorate the final manuscript substantially. I look forward to receiving your revised manuscript.

Reviewer 1 ·

Basic reporting

I feel the manuscript is very well written, and structured appropriately for a scientific paper. There are a lot of references cited, but as indicated below, there are several places in the manuscript where specific citations should be made.

Experimental design

Appropriate, though more comparison among those used would be ideal, in addition to commented on the methods used (versus some of the lethal methods not used) to fully document the state's bee fauna. Overall suitable, though I think the authors need to provide more details in some areas to make the study repeatable.

Validity of the findings

These are adequate, though I think limitations to their approaches should be discussed in more detail.

Additional comments

Review of "Determining Minnesota Bee Species’ Distributions and Phenologies with the Help of Participatory Science"

Overall, an interesting paper in which three methods of survey bee pollinators are used to gain distributional knowledge of bees (iNaturalist, trap nests, non-lethal bumble bee surveys) - the authors provide summaries of their results, including to some extent, the weaknesses of each. However, I think some additional comments on iNaturalist data would be useful...which taxa made up the largest proportiong on non-Research Grade data? Was this because of poor quality images, or due to the inability to provide species-level identification from photos? It seems that iNaturalist plus trap-nests plus bumble bee surveys are likely missing out on some of the really cool bees that are likely not detected, but that would be if pan traps, Malaise traps, or netting (lethal methods) were used, with taxonomists. I really do not feel the authors addressed the limitations to the limits of their chosen methods to really document the species and distributions of all MN bees.

I also think there are several places where definitions and elaboration are needed - the manuscript provided data, a lot of which was provided via the work of non-scientists, though does not necessarily read in all places in a non-scientist friendly manner. For example, the authors introduce the term cleptoparasite, but do not define what that is. I also think there are several instances where references should be inserted (indicated on attached file).

I have also made extensive comments on the attached file.

I would like to see the authors include the species authors (and family) on first mentioned, and feel the authors need to be consistent throughout with regards to using the full binomial name, or abbreviated form (e.g., Bombus affinis versus B. affinis).

Annotated reviews are not available for download in order to protect the identity of reviewers who chose to remain anonymous.

Reviewer 2 ·

Basic reporting

The article is well-written -- good organization, appropriate literature referenced, with good presentation of data.

Experimental design

Research is straightforward and methods described sufficiently.

Validity of the findings

Data are robust. Information provided contributes to knowledge of native bee phenology and distribution in large-scale statewide monitoring effort.

See comments below.

Additional comments

Overall, I thought this was a nice study and paper. I found it easy to read. I was impressed by the three-tiered community science approach, implemented on a statewide level. I think data presented represents an important contribution to our understanding of native bee phenology and distributions within Minnesota. In particular, this study nicely demonstrates the value of community science efforts for monitoring diverse bee communities, including vulnerable and threatened species. I have a few comments and questions regarding the presentation and interpretation of data.

From major to minor questions/comments suggestions:

It’s noted that fewer bumble bees were found in the Prairie Parkland ecological province. It seems that it might be worth further discussing how sampling effort might play into the lower numbers, especially since there were fewer routes run. Note, in methods it says N=6, but only 5 routes seems to appear on the map in Fig. 1. Maybe it’s just hard to see the 6th point. Nonetheless, there is unequal sampling across the three major ecological provinces. The Prairie Parkland province sampling effort is about a third of the effort for the two other provinces. Also, there are fewer iNaturalist observations in western MN in that region, likely due to the lower population densities in those areas.

I presume the higher cropland area would lead to lower abundances as would be expected with habitat loss and degradation due to agricultural intensification. Still, it seem there might be a gap in survey efforts in western MN grasslands. Are there protected natural areas that could be surveyed in that region? Would it be worth considering greater sampling and/or conservation in this region in the future? What do historic distribution data look like? Have you examined GBIF data to explore additional records beyond the UMN insect collection? Also, what was the date range for specimen records from the UMN collection?

Do you think there has been a reduction in the number of bees in the prairie parkland area with land use conversion from prairies to croplands? Is there historical evidence that could be discussed? It seems that in the abstract and discussion there’s some suggestion that this ecoregion might not warrant as much of a conservation focus as the others. With respect to the current distribution of vulnerable bumble bees, especially B. affinis, it does seem to be the case that other ecoregions are better supporting these species. But, I do wonder if that’s because the other habitats have been so degraded.

The loss of prairie/grassland ecosystems throughout MN and North America is a key conservation concern. These habitats are among the most endangered and least protected in the world. In particular, tall grass prairies, which were once common in western MN have largely disappeared. It’s estimated that only 2% of MN prairies are left. I think there should be greater consideration and discussion of how historic land cover changes may influence current species distributions. Also, some caution should be made in using current distributions to establish conservation priorities without the bigger picture context of historic species distributions and land use change. In the future, there might need to be more targeted surveys in protected natural areas in the prairie parkland region and/or conservation efforts to recover habitat and species.

Presumably agricultural intensification in the prairie parkland region has led to a highly fragmented landscape with few intact grassland/prairie habitats… with relevance to Bombus conservation. For example, see: Hemberger et al. 2021:
https://onlinelibrary.wiley.com/doi/abs/10.1111/ele.13786

Thus, these areas may warrant greater conservation efforts. Studies in other states have shown that grassland habitats, including tall grass prairie patches can be important for bumble bees.
For example, see Hines and Hendrix, 2005 Envir. Ent.:
https://academic.oup.com/ee/article/34/6/1477/420911

Other minor questions/comments/suggestions:

Figure 1 legend: Add “Minnesota” or “MN” before “Bee Atlas”  “Locations of Minnesota Bee Atlas observations.”

I think it might be good to change the symbol color & size for nest blocks. It seems that the nest block symbols are larger than the iNaturalist symbols. With the larger size and darker color, they obsure the iNaturalist symbols. While I imagine there will always be some overlap, I recommend using the same size/shape of symbols for the 3 different sampling methods, but different colors. Also, it might be a good idea to pick a different color for the iNaturalist box, since it’s similar to the Tallgrass Aspen Parkland color.

It is stated that: “Megachile relativa, Megachile pugnata, and Heriades carinata, along with their associated parasitic bee species, had distributions that more closely followed the borders separating the LMF, PP, and/or EBF ecological provinces.” There does seem to be an association with particular ecological provinces for these bees, but I’m not sure I see a strong pattern that shows a concentration along the borders. Were statistical analyses conducted to evaluate associations between tunnel-nesting bees and ecological provinces?

In the methods, you indicate that the association between bumble bee abundances and ecological provinces is evaluated, but it seems there were no statistical analyses for the tunnel nesting bees? Is there a reason why? Couldn’t you also determine statistical associations, at least for common species.

Was there any effort to determine the accuracy of the iNaturalist data (i.e., verify identifications)?

Lines: 290-291: Can you provide means +/- SD or SE?

What’s the difference between Fig. 2 and 3? It’s unclear why the figures are distinct and why some species are combined and others are not? In Figure 3, is there a reason that the final map combines three species M. inimical, M. frugalis, and O. georgica, while other rare/uncommon species are mapped alone?

Fig 2 and 3 take up quite a bit of space. I’m not sure that it’s necessary to map out all of the uncommon/rare species (e.g., species with <20 observations), given that you can’t really infer species distributions when the numbers are so low. Could you possibly just focus on species that have sufficient data for distribution maps and then map rare/uncommon species together on a map for supplemental data?

Also, on Figs. 2 and 3 there is symbology not described in the figure legend (i.e. a small flower or circle figure shows up on several maps but is not described).

I was curious to learn more about data from the “stem bundles”. Do you know if bees used stems of all 6 plant species? Was there any evidence that some stems for some plant species were not used or if there were any apparent preferences for certain plant species?

In Figure 4, it’s a hard to see the superscript i in the phenology chart. I actually missed it all together on first read and wondered why iNaturalist data weren’t included in this figure.

Also, what does UMSP stand for in the Fig. 4 table? Presumably that is the UMN Insect Collection? I didn’t see the acronym UMSP used elsewhere or spelled out.

In the earliest possible nest column in Fig. 4 what is the t after 15-May? Check formatting and typos (e.g., italics used in some columns and dates ending with t or to).

Figure 5 Legend: Are all data, including iNaturalist, Bumble Bee Watch, and the “Bumble Bees of North America” for the same period as the bumble bee route data (i.e., 2016-2020) or are the iNaturalist data from 2005-2021 as indicated in Figure 1 legend. That’s not clear from the methods or Figure 5 legend.

It might be good to note in the figure legends the years/dates of data represented in figures, especially when the collection time periods differ.

Please add the reference for the “Bumble Bees of North America” database (e.g., Richardson 2021).

Reviewer 3 ·

Basic reporting

Satyshur et al used a combination of participatory science methods to explore wild bee distributions across different ecological provinces in Minnesota, USA, and potential ecological drivers for some groups. They describe the development and implementation of 3 different projects, varying in their degree of participant engagement, training, and scientific outcomes as a result. Using iNaturalist, a free mobile app for identifying all types of organisms, they expanded the current known distributions of primarily bumble bee species, but also identified new state records, including a non-native species, and expanded distributional knowledge of an endangered bumble bee species. Using artificial bee blocks, they explored both spatial and temporal distributions of cavity-nesting bees. With a structured bumble bee survey along standard routes, they explore landcover metrics driving bumble bee abundances across the state.

The article was clear and well written, with lots of background on methods and their implementation.

Experimental design

Their outline of the different methods and potential outcomes of each were clear in the intro, methods, and well discussed in the discussion. I had a few small questions about analyses and presentation of data, just for clarification, which may need to be addressed (see validity of findings).

Validity of the findings

The implications of their findings are appropriately stated and well discussed in terms of the potential use of participatory science in future monitoring programs and for conservation and management. I did make several notes in need of clarification and also had some questions about data/analysis:

Lines 319: This is an F-test just for the fixed predictor in a linear mixed effect model. What are the other parameter estimates? Year and route were included as random, correct? Shouldn’t those estimates be reported to know how much variation was driven by the random effects?

Line 381: Is there any way to extract what proportion of all observations make it to research grade? This would be comparable to unidentifiable material in traditional specimen -based collections so could hinder research goals. For instance, participants who never get research grade IDs could be put off - could training in photography etc. help?

Figure 2 & 3: I’m assuming that blue symbols for nest blocks <0.1 don’t include average nests/block that were zero? Were there any and if so were they excluded? Just if so, maybe specifically say that in caption?

Also, aren’t figure 2 & 3 the same just displaying distributions of different species? They appear to have the exact same caption which is odd. Should they either be combined as one figure, or explicitly differentiated in the captions, i.e., Figure 2: Osmia, Hylaeus, Heriades, etc. Figure 3: Megachile, etc.

Also, is there a pattern to how the species are laid out? The genera don’t seem to follow an arrangement, with different Genera (e.g. Coelioxys & Megachile in Fig 3) alternating in both vertical and horizontal alignment. Perhaps organize them alphabetically within genus? Or are they laid out by subgenera? Personally, I read multi-panel figure in paragraph form, from top left to bottom right.

Given that these panels include parasitic species reared from other’s nests, maybe highlight that in the caption, since these maps aren’t of Coelioxys nests (as implied by the title ‘tunnel-nesting’, but nests of other species containing parasitic Coelioxys? Clarity could help for unfamiliar readers.

Figure 4: Is ‘UMSP” in the figure synonymous with UMN from the text and caption? Maybe standardize? It’s a little hard to read from the download PDF and I definitely can’t make out the superscripts, even on a giant monitor. Perhaps OK when in print? How big will it be? Maybe alternatively, remove the text columns, which are mostly represented in the colored ranges, and/or superimpose some text or even names over the figure? Or just move to supplemental.

Figure 5. ‘between 1 and 25’, does this mean there were routes with zeros? How were they handled? Curious to see how much effort resulted in zero vs null data. Again, how are these panels organized?

Figures 5 & 6. Why were these figures broken into two panels each, and based on different maximum average abundances, i.e., 1-10 (Fig 5A) , 11-25 (Fig 5B), and 25-180 (Fig 6 A & B, though the differentiation isn’t stated in the caption – no explanation of differences in panels)? Is there a limit to the number of levels you can use for the symbols in the legend categories? Given that the same size symbol in B is 2.5 times more bees on average, this seems to hinder interpretation of relative abundances between species across their distributions.

Figure 7. Does the F-stats need to be redundant here with the results text? Also, why were there < 1/6 as many routes in the prairie region to begin with? There doesn’t appear to be any zeros in this figure. Did no routes find zero bees?

Figure 8. I’d probably report the eigenvalues in the results with other stats and not here in the caption, unless the editor wants them here.

Figure 7 vs Table 1: In the figure there are: 6 PP, ~28 LMF, and ~46 EBF, compared to the table: 6 PP, 18 LMF, and 21 EBF (routes adopted) and 11, 35, and 36 completed, respectively. Why the discrepancy in sample sizes. Ah, because the route by year interaction – year was a random effect. Was the PP route surveyed in only 1 year? Maybe needs a little more explanation in the methods?

Additional comments

Line 428: and M. rotundata. Is it or O. lignaria managed up there for any agriculture?

Line 566: Maybe acknowledge V Scott or A Rose specifically for The Bees Needs?

---

## Round 0.2 · Minor Revisions

Overview
As I believe you have been informed by PeerJ staff, the original Academic Editor was not available to complete the decision on your manuscript. PeerJ asked me to take over responsibility for it after an unsuccessful search for an editor closer to the research topic. I am a retired Behavioral Ecologist with a fairly wide range of experience, including some physiology, morphology and non-behavioral ecology. However, I have never seriously worked on taxonomic issues or invertebrates. Since retirement, I have fairly extensive experience with citizen/participatory science (eBird, some iNaturalist, and various bird-related surveys as well as a short period of participation Bumble Bee Watch). I mention this so that you can understand the problem if you find that some of my comments fail to recognize relevant issues.

I also must apologize for the delay in completing my review. In addition to the time needed to come up to speed with your manuscript, I was about to depart on a short vacation at the time I was asked.

The revised manuscript was reviewed by one of the original reviewers and two new ones. I was not provided an explanation for including two additional reviewers. Two of the reviewers indicated that the responses to previous comments were appropriate and that the article was ready for publication. The third reviewer (Reviewer 5) suggested additional changes. I am very aware of the awkwardness of receiving new criticisms on a revised manuscript. However, I feel that these comments are based on careful reading and thoughtful evaluation and should not be ignored. Although the reviewer categorized the revisions as ‘minor’, they potentially involve new or revised analyses, which are usually considered ‘major’ because they could change results and conclusions. Given this situation, it may be helpful to have my views on the issues raised by this reviewer. I also have a few minor comments based on my reading of the manuscript. I have provided a pdf with a few small grammatical issues noted by highlighting the words of concern and using inserted comments to suggest alternative wording or an explain the problem.

Reviewer 5’s first comment is that the taxonomic experts for iNaturalist provided very substantial contributions and should be more fully recognized in the acknowledgements or even included as authors. I agree with the need for proper recognition but will not insist on co-authorship, given the late stage of the manuscript and the challenge of deciding which identifiers should be considered to have contributed enough to be included as co-authors and then inviting them and waiting for them to confirm their inclusion and to read the manuscript (and possibly request changes). Nevertheless, if you feel that this is a valid concern, you are free to do so as long as they confirm willingness to be co-authors and sign off on the contents of the manuscript.

Reviewer 5’s second point is that bumble bee abundance may reflect multiple individuals from the same colony and an analysis based on presence/absence might be more appropriate. This would affect your analysis of the association between bumble bees and land use. I presume that similar concerns would affect any quantitative survey of social insect abundance because number of colonies and individuals per colony would both influence abundance. If abundance surveys are generally accepted in studies of social insect abundance, I would be comfortable with a reference to that effect and including in the Discussion any potential biases that might result. You must judge how much of a concern this is and whether you should carry out a presence/absence analysis to see if it alters conclusions, perhaps included in supplemental material.

This reviewer’s third point also involves a potential major reanalysis because they propose that the landscape analysis be based on the specific survey locations rather than the center of the route. The point that the route center may not reflect local habitat throughout the route is a valid one, of course. Again, there may be precedent for your approach, if not for bees then perhaps in research based on Breeding Bird Surveys. This seems to me a very large request at a late stage of manuscript production for a relatively small part of the study (one sentence in the Abstract) and I would be comfortable with an addition to the Discussion, raising the issue and indicating how it might affect conclusions. For example, routes may have been chosen to be relatively homogenous in land use, which would reduce this concern.

Most of Reviewer 5’s other comments are useful suggestions that relate to clarifications of the text. However, their point regarding L255 proposes a change in how the independent variable ‘year’ is included in the analysis. I do not have the expertise to judge this; you should consult a statistician to decide whether a different analysis is more appropriate. Two other points in which the reviewer is not wrong but I do not necessarily agree with the need to change are the following:
• I do not agree that you need to move the text information sections starting on L331 and L408 to tables though you may do so if you agree with the reviewer.
• The sentence starting on L642 can be left in to provide a fuller context.

Specific suggestions from the editor
L121. Here, and a number of other places, genera are not italicized. Please check the entire text carefully.
L129. The statement that the female builds a nest and then plugs the tunnel entrance could be clarified by mentioning provisioning and egg laying, I think.
L317 and elsewhere. I am not familiar with a style that uses a comma in numbers with five digits but not four. I don’t recall that PeerJ requests this style. In general, I would expect commas in any number greater than 999. Is there a good justification for your decision? I highlighted other four-digit numbers missing a comma but may not have noted all of them.
L569. I think that ‘cow chips’ is a slang or colloquial term that may not be understood by readers for whom English is not their first language (and maybe some for whom it is!). Could you replace with something like ‘dried cattle feces’?
L592. I think you need to provide start and end years for a 10.4-day change in phenology to have any meaning to readers.
L721ff. The references need to be checked. There are some missing italics in genera and species. There is inconsistent use of capital letters in journal titles. A few titles are in italics, but I could not see why. I highlighted some examples but did not carefully review the entire section.
The caption to Fig. 3 is highly redundant to Fig. 2. See if you can remove the repeated details from the Fig. 3 caption and refer the reader to Fig. 2.
Similarly, consider reducing the redundancy in the caption to Fig. 7 by referring to Fig. 6.

Reviewer 3 ·

Basic reporting

As before, the article was clear and well written, with lots of background on methods and their implementation and justification for edits, etc. in the response to reviewers.

Experimental design

Additional details provided in the revised manuscript help differentiate the different methods, results, and add context.

Validity of the findings

I think the implications of their findings are appropriately stated and discussed, especially in terms of the potential use (and limitations) of participatory science in future monitoring programs and for both conservation and management.

Additional comments

The authors have thoroughly addressed my suggested revisions.

Reviewer 4 ·

Basic reporting

The paper was very well written and with extensive citation to contextualize the research objectives. The project outlines a multi-year bee inventory project, one of the most productive in the U.S., and the objectives of this survey are clearly articulated in the last paragraph of the introduction.

Experimental design

The paper reports on multiple well-established methods for bee survey. The scale of this survey (on the level of a state) and the high level of reliance on volunteers, makes this contribution significant.

Validity of the findings

There are certainly some limitations associated with the relatively low intensity of survey methods used in this study. But the authors clearly articulate the reasons for this level of survey, namely the listing of a bumble bee species under the Endangered Species Act, which prevented higher intensity passive trapping methods. The authors, however, clearly acknowledge these limitations in the discussion section.

Additional comments

Many states are searching for a model to conduct native bee surveys at a state-level. This is the first comprehensive survey of bee fauna at such a broad scale that I am aware of. While I do have concerns about the capacity of some of the methods to detect rare species, the authors do a good job of contextualizing their findings and discussing the trade-offs they faced when developing their protocols.

Reviewer 5 ·

Basic reporting

[Clear and unambiguous, professional English used throughout] Yes, the manuscript is very well written.

[Literature references, sufficient field background/context provided] Yes

[Professional article structure, figures, tables. Raw data shared] Yes

[Self-contained with relevant results to hypotheses] Yes

Experimental design

[Original primary research within Aims and Scope of the journal] Yes

[Research question well defined, relevant & meaningful. It is stated how research fills an identified knowledge gap] Yes

[Rigorous investigation performed to a high technical & ethical standard] Yes, but I believe that there are some issues with statistical analysis choices

[Methods described with sufficient detail & information to replicate] Yes

Validity of the findings

[Impact and novelty not assessed. Meaningful replication encouraged where rationale & benefit to literature is clearly stated] Not applicable, study is novel.

[All underlying data have been provided; they are robust, statistically sound, & controlled] Yes

[Conclusions are well stated, linked to original research question & limited to supporting results] Yes, though some additional details would be appreciated (see additional comments).

Additional comments

The authors present a nice set of data regarding contributions of participatory science to inventorying and monitoring bee diversity, distribution, and natural history for Minnesota, on the heels of a related paper reporting the checklist of Minnesota bees. This revised manuscript is well written and well organized, and the authors have systematically addressed the numerous issues brought up by previous reviewers. This study is an excellent contribution to the field and will be an important case study with respect to the partnerships between professional biologists and the lay public in bee conservation. Nevertheless, perhaps due to the longer length of the manuscript, I do have a number of major and minor comments that I believe would be necessary for the authors to satisfactorily address before this work is published.

Major comments:

1. The authors mention in L86-87 that they relied on “crowd-sourced identifications” to “efficiently [provide] presence data”, and cursorily mention several leading bee experts (L122) as trustworthy ID providers. In the case of bee identification via photography, as the authors themselves point out explicitly in L122, true “crowd-sourcing” is untrustworthy and thus data quality really rests upon participation by true experts—who in most cases are professional scientists. However, these scientists who have expended enormous effort to shape the quality and usability of the iNaturalist dataset were not even included in the acknowledgements alongside others who provided specimen IDs (sometimes for a single species). I would even venture to say that if the IDs pf these iNaturalist-enhancing scientists have been instrumental to the success of this particular methodology, the appropriate course would be approaching them to discuss coauthorship as one would in the case of taxonomists who provided IDs for large portions of physical specimens collected for a project, given their very material participation in making this study possible (not being able to clearly evaluate the exact contribution of all ID providers should not prevent acknowledging / involving at least the most obvious and important contributors). As it stands the manuscript mentions engaging “volunteers” and “users” but to me it seems to me like on the iNaturalist front, the authors refer only to the observers, whereas in actuality the ID providers are just as much “volunteers” and “users”, but did not receive engagement from the authors (if there was engagement, it was not reported). Lastly, L704 mentions “professional experts”, so it does seem like the authors recognize the importance of such individuals—but it does not seem like the authors have considered these ID providers to belong to this category with this mention in L704 (they were never referred to as such).

2. A previous reviewer had brought up the point that bumble bee individual abundance at a given place and time may not be relevant, since it could be strongly influenced by allocation of individuals from the same colony. The authors responded that they only used iNaturalist data for presence-only analysis, which is fair enough—but here, they still used observed bumble bee abundance as a variable for the bumble bee-specific protocol (e.g., L253), which I think falls under the same criticism. At the minimum I believe that it should be necessary to do an auxiliary analysis where bumble bee incidence (presence vs absence at a given stop, combined across all times of year), rather than abundance, is used as the response variable; and report these results at least in an appendix. In this analysis, I think stops on the same route that are too close to one another (maybe within 2 km?) should be combined into a single data point, again due to the possibility of counting individuals from the same colony (this shouldn’t cause too many to be combined, I’m guessing, given the average separation of > 5 km among sites).


3. Again regarding the bumble bee protocol, specifically L266-267: If the routes are nearly 40 km long, I imagine a lot can change within this space, and some large portion (or perhaps all) of the stopping points where bumble bees are counted could have vastly different local- or landscape-scale conditions compared to the route central point where land cover is analyzed—in such cases a land cover analysis based only on the center point of the route would be completely uninformative. It seems to me that if some kind of landscape analysis is to be done, it is necessary that it be done at the specific locations where the observer stopped to look for bees (i.e., up to 15 locations per route if every run had unique stopping locations), or, along the entire route (i.e., a buffer radiating from the entire linear route).


Specific / minor comments:

L37: I would encourage the authors to consider “volunteers” and iNaturalist users not only those who provide data, but also those who provide identifications (see also major comment #1 above)

L59-60: Please cite a reference for these ecological provinces. How do they relate to ecoregions?

L108: Is this 2300 volunteers who were specifically recruited for this project? Or does some portion of this number consist of naturalist hobbyists who would’ve submitted observations anyway (including veteran users of the platform), who may not have even heard of this project? It would be good to specify. From the way the manuscript is written (e.g., L81-82) it sounds like the authors personally interacted with these 2300 iNaturalist users. If that is not the case, then it does not seem appropriate to present it as having “engaged” these “volunteers” (L81) if what was done is simply mining the observations of otherwise not-contacted observers (and more seriously, ID providers, see also major comment #1 above).

L122: If these experts identified a sizable portion of the data, I believe that they (and others who routinely offer high-quality IDs such as Joel Neylon @neylon, Tony Ernst @gamelaner, and Brian Dagley @bdagley) should be explicitly mentioned in the acknowledgments at the very bare minimum (please see also major comment #1 above).

L124-125: Did all 2300 volunteers (L108) attend workshops? It would be good to know how many workshops were held and how many volunteers attended.

L185: By “county center” does this mean centroid?

L189: It would be nice to have some information about the criteria used to judge quality.

L192-194: If the volunteers are reporting truly “empty” tunnels, I would think that it would be fair to assume that no bee can complete its entire series of nests cells in a single day, or even 2-3 days (there are published studies that have investigated these dynamics) so it seems that one can shave off several days from this timespan, based on the taxon’s (genus or subgenus?) published nest completion rates? Unless of course, if by “empty” it just means “uncapped”, in which case the approach here is likely the most valid, since “empty” could mean that the bee/wasp is just one cell away from completion and can complete it that same day.

L227-228: The clause “as listed by the International Union for the Conservation of Nature” seems misplaced as it appears to refer specifically to the Psithyrus as currently written; I doubt that is the case?

L231: What does it mean when an observation is unverifiable in this case? That a volunteer indicated that they don’t know the species AND did not take a photograph? Or that there is a photo and EE was unable to discern species based on the image submitted? Please clarify.

L232-233: Does this grouping constitute the 1% of unverifiable bumble bees? Or is the 1% unverifiable apart from this group?

L243: Negative binomial what (distribution)?

L255: I strongly disagree that singularity should be handled by simply dismissing a potentially important variable. Year is probably borderline with respect to its suitability as a random effect anyway since there are temporal autocorrelations among years (abundances of one year may be tied to those of the preceding and following year). I would encourage the authors to think about alternative ways to construct the model while still meaningfully accounting for year. One common approach is to simply include it as a fixed-effect covariate.

L261-262: Did the authors examine aerial imagery for all relevant areas? Or a subset? If subset, how was the subset chosen?

L304: These are notably absent among records determined to the species level, not absent from iNaturalist altogether—though the difference is subtle, I think it is important to make this distinction, since better AI and/or growing experience of taxonomist making IDs via photos may ultimately be able to discern many of these currently “absent” species.

L331-362: This is all very interesting and useful information. I find it not very easy to navigate when it’s just a block of text, though, and wonder if these results can be better displayed as one or multiple tables? Perhaps results reported in L408-421 fall in this category as well.

L481-482: I disagree with the sentiment that “iNaturalist records alone should not be used to describe the structure of the bee community as they do not provide a complete view of species diversity.” Given that many community-level structural properties can be described and assessed at least at the coarse scale based on genus-level identifications, I think iNaturalist data can VERY well serve to describe bee community structure, just not the species-level taxonomic composition and measures that necessitate this level of resolution.

L491: Please clarify what is meant by “doing particularly well”. High individual health condition? High fecundity? Large local population size?

L517-521: Interesting discussion. But, main subject here—Heriades carinata—is generalist as far as I recall and should therefore experience more limitation by nesting substrate (in this case resin) by this train of logic??

L547-548: I was surprised that this caveat wasn’t one of the first to be mentioned, given its likely importance in influencing the data. Figures 2 and 3 seem to be able to lend some insight into this particular challenge?

L634-635: Do we REALLY want to improve roadside habitat though, since there are excellent patches of habitat away from roadsides in this province? I understand that roadsides represent excellent and underutilized opportunities, but all else being equal, restoring non-roadside habitat has the benefit of NOT increasing mortality via vehicle collisions and road pollution.

L642-645: This repeats the results section and could be abbreviated?

L646-648: It would be great if the authors could mention what these “new insights” are!

L650-661: It sounds like there are opportunities here to cross-reference bumble bee route surveys and iNaturalist results to see how these two methods compare to each other, not just for B. affinis but all of the other species too. Is there a reason this was not done?

Table 6: I’m not quite understanding the stats being reported here—what is the reference level and what does the P value refer to? E.g., LMF has no significant effect on what, and relative to what? What does it mean to have a significant effect here? The Methods and Results sections could use a bit more clarity in describing exactly what was done to generate these results.

---

## Round 0.3 · accepted · Accept

The authors have produced a careful and thorough response to the reviewer's recommendations. I believe the manuscript to be ready for publication.